# GEM-Bench: A Benchmark for Ad-Injected Response Generation within Generative Engine Marketing

## Abstract

Generative Engine Marketing (GEM) is an emerging ecosystem for monetizing generative engines, such as LLM-based chatbots, by seamlessly integrating relevant advertisements into their responses. At the core of GEM lies the generation and evaluation of ad-injected responses. However, existing benchmarks are not specifically designed for this purpose, which limits future research. To address this gap, we propose GEM-Bench, the first comprehensive benchmark for ad-injected response generation in GEM. GEM-Bench includes three curated datasets covering both chatbot and search scenarios, a metric ontology that captures multiple dimensions of user satisfaction and engagement, and several baseline solutions implemented within an extensible multi-agent framework. Our preliminary results indicate that, while simple prompt-based methods achieve reasonable engagement such as click-through rate, they often reduce user satisfaction. In contrast, approaches that insert ads based on pre-generated ad-free responses help mitigate this issue but introduce additional overhead. These findings highlight the need for future research on designing more effective and efficient solutions for generating ad-injected responses in GEM.

## 1 Introduction

Large Language Models (LLMs) and general-purpose AI assistants, such as ChatGPT and Gemini, are fundamentally reshaping how users interact with information by providing direct, conversational, and context-aware answers. This shift is disrupting traditional search engine marketing (SEM) that relies on clicking sponsored page on the search engine. Notably, Brodsky (2023) reports that 35% of casual users find LLMs more helpful for obtaining information than search engines. A recent study (Chapekis & Lieb, 2025) confirms that Google users' click-through rate (CTR) is 8% when AI summaries are present, compared to 15% without them, representing nearly a 47% decline in CTR on traditional links when AI summaries appear. Meanwhile, to monetize LLM models, service providers such as Bing Chat (Sainsbury-Carter, 2023) and Perplexity (Team, 2024) have begun experimenting with integrating advertisements directly within LLM-generated responses, referred to as *generative engine marketing* (GEM).

This work focuses on a linchpin problem of GEM, called the ad-injected response (AIR) generation. Given an advertisement database $\mathcal{D}$ and an LLM $\mathcal{M}$, the AIR generation problem takes as input a user query and optional context (e.g., user profile or chat history) and aims to seamlessly integrate relevant ad descriptions from $\mathcal{D}$ into the $\mathcal{M}$-generated response to this query. As a new research direction, datasets, evaluation metrics, and solutions for AIR generation remain limited. Existing LLM benchmark datasets (Hendrycks et al., 2020; 2021; Zheng et al., 2023b;a) contain many questions from domains that are fundamentally unsuitable for ad integration, such as mathematics, coding, and data labeling. Moreover, common text quality metrics such as ROUGE (Lin, 2004) and BLEU (Papineni et al., 2002) evaluate responses against reference texts, which introduces bias when ads are injected into pre-generated templates or responses. At the same time, recent LLM-as-a-Judge approaches (Zheng et al., 2023b;a) are not tailored for evaluating AIR, since they fail to capture nuanced aspects of user experience such as trust or jump-out feeling (Tang et al., 2024). Regarding existing solutions, Feizi et al. (2023) propose the first conceptual framework for LLM-based advertising. In this framework, the authors focus on formulating the stages of this emerging

advertising ecosystem. Although they mention the idea of first generating an ad-free response and then modifying it by inserting suitable ads for AIR generation, their main focus is on the broader mechanism design, e.g., whether the platform or the advertiser performs this modification and the associated challenges, whereas the specific strategy for how to perform the modification remains an open question. Tang et al. (2024) design a chatbot for AIR generation in which relevant ads are inserted into the system prompt to personalize responses and promote products or brands, yet its performance has not been extensively studied.

To address these issues, we introduce GEM-BENCH, the first comprehensive benchmark for AIR generation in GEM. GEM-BENCH consists of curated datasets, a measurement ontology, and several vanilla solutions for evaluating AIR generation in GEM. Specifically, we curate two datasets, MT-Human and LM-Market, for chatbot scenarios, containing user queries to ChatGPT that are suitable for ad injection, and one dataset, CA-Prod, that simulates the AI overview feature in search engines. To evaluate the quality of AIR, we propose a metric ontology that captures multiple aspects of user satisfaction and engagement. This ontology can also be applied to other stages of GEM, such as estimating response quality during bidding and auction. Finally, we propose a multi-agent paradigm, called Ad-LLM, for AIR generation, which first produces ad-free responses and subsequently injects advertisements with refinement. It decouples agents and workflows, enabling flexible development of future solutions. We also instantiate several baseline solutions under this paradigm. Our preliminary experiments reveal clear trade-offs among different solutions. We find that the simple system prompt-based approach (Tang et al., 2024), while achieving relatively high engagement (e.g., 6.5% higher CTR), often compromises user satisfaction by reducing response accuracy and trust. In contrast, Ad-LLM improves user satisfaction and engagement, achieving relative gains of up to 39.8% in accuracy, 89.3% in personality, and 62.9% in trust across multiple datasets. These improvements are consistently validated by different LLMs and human participants, reaching a strong consensus. However, this baseline incurs additional overhead, consuming about $2\times$ more output tokens than the system prompt-based solution.

To summarize, we make the following contributions in this work:

- We introduce GEM-BENCH, the first comprehensive benchmark for evaluating ad-injected response (AIR) generation in generative engine marketing (GEM).
- We curate and preprocess three datasets from two real-world scenarios, chatbots and search engines, where AIR generation can be effectively applied.
- We design an evaluation ontology for assessing the quality of AIR, considering multiple aspects of user satisfaction and engagement.
- We introduce a multi-agent framework Ad-LLM with several baseline implementations for AIR generation, enabling researchers to easily experiment with and develop new solutions.

## 2    PRELIMINARIES

This section first introduces the existing frameworks for search engine marketing (SEM) and generative engine marketing (GEM), followed by formulating the Ad-Injected Response (AIR) Generation problem and briefly reviewing the existing solution, named Ad-Chat (Tang et al., 2024). Additional related work is summarized in Appendix A.

### 2.1    SEM AND GEM

The online advertising ecosystem consists of three stakeholders: platforms, users, and advertisers. *Platforms* connect advertisers with users by delivering targeted ads, aiming to balance user experience with sustainable monetization. *Users* consume content and may interact with ads, expecting minimal disruption. *Advertisers* pay for placements to reach target audiences and drive conversions.

**Search Engine Marketing (SEM).** As a primary channel of online advertising, SEM displays sponsored results in response to user queries on a search engine. The standard paradigm (Lahaie et al., 2007) typically consists of the following core stages:

- *Stage 1: Offline bidding.* Advertisers first configure their campaigns by placing monetary bids on one or more keywords relevant to their ads.

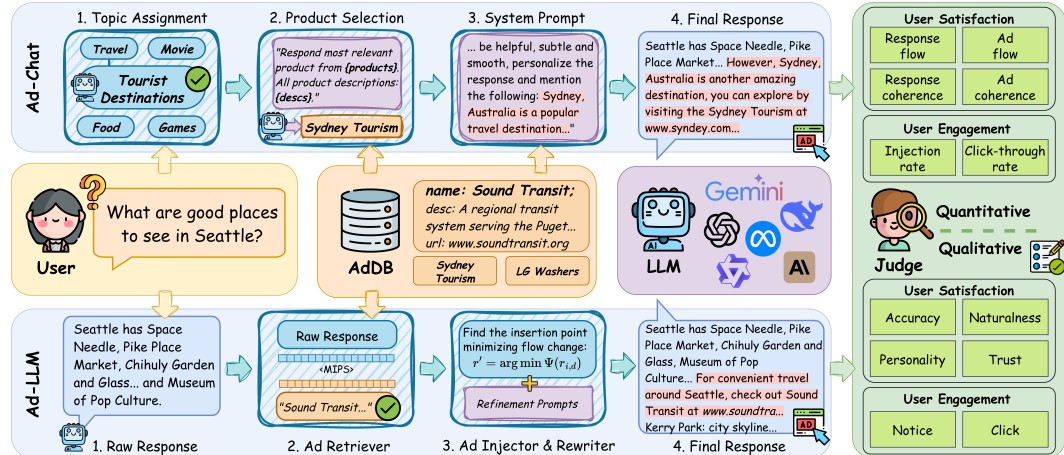

Figure 1: The illustration of two existing workflows for ad-injected response (AIR) generation in GEM and the proposed measurement ontologies.

- *Stage 2: Query processing.* When a user submits a query, the platform retrieves a ranked list of organic results and, in parallel, collects a set of candidate ads that match the query.
- *Stage 3: Slot identification.* Based on the query and result list, the platform determines how many ad slots will be displayed.
- *Stage 4: Performance prediction.* For each advertiser–slot pair in the candidate set, the platform predicts the likelihood of a user clicking the ad, i.e., the click-through rate (CTR).
- *Stage 5: Auction and allocation.* Finally, the platform runs an auction to allocate ads across the available slots, aiming to maximize overall expected value by considering bid amounts and CTR.

**Generative Engine Marketing (GEM).** Analogous to SEM, Feizi et al. (2023) propose a conceptual framework for generative engine marketing (GEM), in which an LLM integrates native advertisements directly into its generated responses. This framework mirrors the five stages of SEM but extends them to the context of LLMs. In particular, during query processing (stage 2), the system first generates an ad-free response to the user query and then performs an additional modification step to inject a relevant ad candidate into the output.

Although the overall pipeline can be adapted from SEM, each stage of GEM introduces unique challenges and open questions. In the bidding stage, a central issue is determining what exactly advertisers would bid for. In the slot identification stage, the absence of explicit ad slots raises two key problems: deciding whether a query is suitable for ad inclusion and, if so, identifying insertion points within unstructured text that may contain multiple candidates. For performance prediction and auction, a fundamental challenge is estimating click-through rates (CTR) from unstructured responses. Beyond explicit engagement signals, the system must also evaluate and predict user satisfaction with the generated content. Finally, in the ad-injected response (AIR) generation stage, the central problem is how to produce output that *(i) incorporates ads seamlessly without degrading the user experience, and (ii) satisfies the advertiser's objectives.*

## 2.2 AD-INJECTED RESPONSE GENERATION

In this work, we focus on the Ad-Injected Response (AIR) Generation problem. It is the linchpin step of GEM, since regardless of how the overall framework evolves, the AIR is eventually what the system delivers to the user.

**Formulation.** We represent the results of the offline bidding stage as an advertisement database (AdDB) $\mathcal{D}$, where each row $d_i \in \mathcal{D}$ contains the bidding string, its associated embedding, bidding price, and product meta-information (such as advertiser, description, URL, and ad creatives). Given an AdDB $\mathcal{D}$, an LLM $\mathcal{M}$, and an integer $k$, the AIR generation problem takes as input a user query and possible context (e.g., user profile or chat history) and aims to return an ad-injected response (AIR) that seamlessly integrates $k$ unstructured and relevant ads from $\mathcal{D}$ into the $\mathcal{M}$-generated response to the query. The objective is to maximize both user satisfaction and user engagement. Satisfaction reflects the user experience when interacting with the model $\mathcal{M}$ and can be measured

Table 1: Statistics of curated datasets.

| Dataset | Scenario | AdDB | | | Queries | | |
|---|---|---|---|---|---|---|---|
| | | #Ads | #Topics | Source | #Queries | #Topics | Source |
| **MT-Human** | Chatbot | 6,556 | 576 | (Tang et al., 2024) | 10 | 1 | (Zheng et al., 2023b) |
| **LM-Market** | Chatbot | 6,556 | 576 | (Tang et al., 2024) | 1,701 | 3 | (Zheng et al., 2023a) |
| **CA-Prod** | Search Engine | 2,215 | 6 | (Zhu et al., 2022) | 120 | 6 | (Zhu et al., 2022) |

by factors such as response latency and content quality. Engagement, on the other hand, captures whether the user notices the ad, has a positive attitude toward it, and potentially clicks the in-text ad link, which are the dimensions of primary interest to advertisers. For simplicity, we assume that every query discussed here has already passed the slot identification stage and is therefore suitable for ad insertion, and we set $k = 1$ in the subsequent discussion.

**Existing Solution: `Ad-Chat`.** Beyond the idea of inserting a relevant ad into an original response (Feizi et al., 2023), Tang et al. (2024) propose `Ad-Chat`, which integrates native ads into the system prompt when responding to user queries. As illustrated in Figure 1, given a user query and possible user profile and chat history, `Ad-Chat` first uses the LLM $\mathcal{M}$ to assign a topic from Google Topics (Google Privacy Sandbox Team, 2024) to the conversation. It then instructs $\mathcal{M}$ to select the most suitable product from the bidding products associated with that topic. Finally, `Ad-Chat` inserts the ad with its description and URL into the system prompt, aiming to appeal to the user and personalize the response while promoting products, ultimately generating the AIR.

## 3 GEM-BENCH

This section introduces the main components of GEM-BENCH: curated datasets for both chatbot and search engine scenarios, a measurement ontology for evaluating AIR, and a modular framework implementation. Detailed descriptions of each component are provided in Appendices B, C, and D.

### 3.1 DATASETS

**Chatbot: MT-Human and LM-Market.** For the chatbot scenario, we construct two query sets. MT-Human is curated from MT-Bench (Zheng et al., 2023b), where each query is manually examined for suitability for ad insertion. We retain all 10 first-turn queries from the humanities category to form the MT-Human dataset, including examples such as "What are some business etiquette norms when doing business in Japan." Due to the large scale of LMSYS-Chat-1M (Zheng et al., 2023a), LM-Market is derived through a multi-stage pipeline: we filter for single-turn English queries, cluster them into topics, and then apply LLM-assisted scoring followed by manual verification to retain queries with high ad-injection potential. In total, 1,701 queries are selected across three domains: *travel planning*, *recipe recommendation*, and *software tools comparison*. The associated AdDB $\mathcal{D}$ is adopted from `Ad-Chat` (Tang et al., 2024), which contains 6,556 manually verified products, brands, and organizations spanning 25 main topics and 576 subtopics.

**Search Engine: CA-Prod.** For the search engine scenario, we construct CA-Prod from a commercial dataset of query–ad pairs with human-annotated relevance labels (Zhu et al., 2022). Following a procedure similar to LM-Market, we cluster ads into six representative domains, such as *nutrition supplements*, *women's dresses*, and *Android devices*, and assign topics to queries according to their relevant ads. After filtering out extremely sparse or dense cases, we retain 120 queries associated with 2,215 unique products. To mimic the AI overview feature in modern search engines, the inference model $\mathcal{M}$ is set to generate a concise overview for each keyword query, helping users quickly grasp the topic. The statistics of the curated datasets are summarized in Table 1.

### 3.2 MEASUREMENT ONTOLOGY

**Quantitative Metrics.** Consider a response $r$ with $\ell = |r| > 1$ sentences, where $s_i$ denotes the $i$-th sentence and $\mathbf{s}_i$ its embedding. Let $\text{sim}(\cdot, \cdot)$ denote cosine similarity, and $\bar{\mathbf{s}}$ the mean embedding of all sentences. To capture user satisfaction, we define two global metrics *response flow* and *response coherence*, and two corresponding local metrics *ad flow* and *ad coherence*. Specifically, *response*

*flow* measures semantic continuity across adjacent sentences:

$$f(r) = \frac{1}{\ell-1} \sum_{i=1}^{\ell-1} \text{sim}(\mathbf{s}_i, \mathbf{s}_{i+1}),$$

while *response coherence* evaluates how consistently each sentence aligns with the overall theme:

$$c(r) = \frac{1}{\ell} \sum_{i=1}^{\ell} \text{sim}(\mathbf{s}_i, \bar{\mathbf{s}}).$$

*Ad flow* measures whether semantic transitions around an injected ad remain smooth, and *ad coherence* captures how well the ad aligns with the non-ad content. For engagement, we adopt *injection rate* to record whether a response contains an ad, and *click-through rate* to estimate further user conversion.

**Qualitative Metrics.** Beyond similarity-based measures, we include six LLM-as-a-Judge metrics that capture user satisfaction by *accuracy*, *naturalness*, *personality*, and *trust*, and user engagement by *notice* and *click*. Each metric is evaluated along two dimensions. Accuracy evaluates whether the response directly answers the user's question based on relevance and factual correctness. Naturalness measures how ad insertion affects conversational flow through interruptiveness and authenticity impact. Personality assesses whether the assistant remains user-centered by examining helpfulness and the degree of sales-driven behavior. Trust evaluates users' confidence in the response by assessing holistic credibility and perceived bias introduced by ads. Notice simulates whether users recognize mentioned products using product awareness and their resulting attitude. Click estimates link-level engagement using link awareness and the likelihood of clicking based on relevance and appeal. For each dimension, the judge assigns one of three quality levels: bad, moderate, or good. These two dimension ratings are then aggregated into the final discrete score using a unified rubric: 0 for both dimensions rated bad; 30 for one bad and one moderate; 60 for both moderate or a combination of bad and good; and 90 for both good. This explicit rubric, together with dimensions grounded in a codebook from real user studies (Tang et al., 2024), ensures both validity and practical relevance.

### 3.3 Ad-LLM Framework

We introduce Ad-LLM, a modular multi-agent framework for AIR generation. As shown in Figure 1, it follows a chain-like workflow with four agents: an ad-free response generator, an ad retriever, an ad injector, and a response rewriter. Given a query $q$, an ad database $\mathcal{D}$, and an LLM $\mathcal{M}$, the framework operates as follows. The response generator first produces an ad-free response $r$. The ad retriever then selects a small set $\mathcal{D}_t$ of relevant ads by measuring semantic similarity between either the query or the generated response and the ad contents. The ad injector determines both which ad to insert and where to place it by minimizing the *flow disturbance* introduced by the insertion. For a response $r$ with $\ell$ sentences, inserting an ad sentence $s_d$ after $s_i$ yields $r_{i,d}$. Given retrieved candidates $\mathcal{D}_t$, we choose

$$r' = \arg\min_{i \in [\ell-1],\, d \in \mathcal{D}_t} \Psi(r_{i,d}),$$

where

$$\Psi(r_{i,d}) = \text{sim}(\mathbf{s}_i, \mathbf{s}_{i+1}) - \frac{\text{sim}(\mathbf{s}_i, \mathbf{s}_d) + \text{sim}(\mathbf{s}_d, \mathbf{s}_{i+1})}{2}.$$

This objective quantifies the disturbance at the injection point and selects the ad–position pair that least disrupts the original flow. The response rewriter then refines $r'$ using $\mathcal{M}$ to ensure that the injected ad integrates smoothly with the surrounding context. Notably, Ad-LLM also supports multi-$k$ insertion. After inserting the best ad according to the disturbance objective, the updated response becomes the input for the next selection and insertion step. This iterative procedure continues until $k$ ads have been inserted, applying the same criterion at each iteration.

## 4 Experiments

GEM-BENCH aims to answer the following three research questions:

- **RQ1**: How effective are solutions in terms of quantitative and qualitative evaluation ontologies?
- **RQ2**: How do humans react to AIR, and how well their evaluations align with LLMs?
- **RQ3**: How do the core agents and their implementations impact the effectiveness of Ad-LLM?
- **RQ4**: How much does each solution cost in terms of waiting time and monetary cost?

Table 2: Quantitative effectiveness evaluation. RF: Response Flow, RC: Response Coherence, AF: Ad Flow, AC: Ad Coherence, IR: Injection Rate, CTR: Click-Through Rate.

| Dataset | Solution | Quantitative Metrics | | | | | | |
|---|---|---|---|---|---|---|---|---|
| | | RF | RC | AF | AC | IR | CTR | Overall |
| MT-Human | Ad-Chat | *82.06* | *41.12* | **43.48** | *62.39* | 66.00 | — | 59.01 |
| | GI-R | **87.06** | **41.70** | 43.10 | **66.32** | 100.00 | — | **67.64** |
| | GIR-R | 77.08 | 39.95 | 41.77 | 61.67 | 100.00 | — | 64.09 |
| | GIR-P | 77.60 | 40.40 | 41.56 | 61.72 | 100.00 | — | *64.25* |
| LM-Market | Ad-Chat | *82.26* | **54.30** | *42.96* | *66.18* | 95.82 | — | *68.30* |
| | GI-R | **84.09** | 50.38 | **44.37** | **68.81** | 100.00 | — | **69.53** |
| | GIR-R | 73.35 | *51.23* | 42.53 | 65.60 | 100.00 | — | 66.54 |
| | GIR-P | 74.38 | 49.95 | 42.15 | 65.43 | 100.00 | — | 66.38 |
| CA-Prod | Ad-Chat | 85.92 | 43.52 | 35.99 | 65.34 | 100.00 | **42.02** | 62.13 |
| | GI-R | **86.12** | **63.49** | 42.10 | **69.46** | 100.00 | 35.23 | **66.07** |
| | GIR-R | 81.07 | *62.58* | **43.09** | 66.83 | 100.00 | 35.23 | 64.80 |
| | GIR-P | 77.87 | 61.99 | *42.92* | *67.12* | 100.00 | *39.45* | *64.89* |

Table 3: Effectiveness evaluation in terms of qualitative ontology.

| Dataset | Solution | Qualitative Metrics | | | | | | |
|---|---|---|---|---|---|---|---|---|
| | | Accuracy | Naturalness | Personality | Trust | Notice | Click | Overall |
| MT-Human | Ad-Chat | 72.60 | 51.00 | 68.40 | 61.80 | 72.60 | 64.20 | 65.10 |
| | GI-R | 83.40 | 39.60 | **79.80** | 68.40 | 74.40 | 72.60 | 69.70 |
| | GIR-R | **85.80** | *52.80* | 74.40 | **73.20** | **82.80** | *79.20* | **74.70** |
| | GIR-P | *82.80* | **53.40** | *74.40* | 70.20 | *81.60* | **81.00** | *73.90* |
| LM-Market | Ad-Chat | 61.97 | 52.54 | 57.38 | 55.47 | 77.46 | **78.01** | 63.81 |
| | GI-R | *79.55* | 47.46 | **76.49** | 69.30 | 75.11 | 74.84 | 70.46 |
| | GIR-R | **80.32** | **62.49** | *71.88* | **70.47** | **80.14** | *77.10* | **73.73** |
| | GIR-P | 78.96 | *60.72* | 71.63 | *69.48* | *79.66* | 75.52 | *72.66* |
| CA-Prod | Ad-Chat | 42.60 | **36.38** | 25.03 | 22.66 | **88.12** | **88.25** | 50.51 |
| | GI-R | 53.76 | 26.55 | 44.85 | 36.69 | 80.09 | 85.08 | 54.50 |
| | GIR-R | **59.56** | 34.45 | *46.97* | **36.92** | *84.97* | 87.16 | *58.34* |
| | GIR-P | *59.19* | 35.40 | **47.39** | *36.88* | 84.34 | *87.24* | **58.41** |

GEM-BENCH evaluates the performance of four solutions: the existing solution Ad-Chat (Tang et al., 2024) and three instances, GI-R, GIR-R, and GIR-P, of the implemented framework Ad-LLM across the aforementioned datasets and evaluation metrics. For LM-Market, we sample 100 queries from the three domains above for evaluation. We additionally scale the dataset to 1,000 queries to assess robustness, and the results remain consistent (see Appendix E). For the three Ad-LLM instances, GIR represents completing all three agents to Generate, Inject, and Rewrite, but GI skips the final rewriting step. The suffixes -P and -R indicate whether the ads are retrieved based on the user prompt or the original response, respectively. Unless otherwise specified, all solutions use doubao-1.5-lite-32k as the base LLM, and their results are evaluated using gpt-4.1-mini. Additionally, for quantitative evaluation and Ad-LLM instances, we set the embedding model to text-embedding-3-small by default. For Ad-Chat, we follow its original implementation and retain the default configuration, including the system prompts. In addition, we examine the impact of varying system prompts and show in Appendix F that the performance is insensitive to these prompt changes.

## 4.1 EFFECTIVENESS EVALUATION (RQ1)

**Ad-LLM outperforms Ad-Chat in user satisfaction and user engagement.** Table 2 and Table 3 report the performance of Ad-Chat and three Ad-LLM instances in terms of both quantitative and qualitative evaluation ontologies across three datasets. Overall, the best Ad-LLM instance improves Ad-Chat by 14.6%, 1.8%, and 6.3% in terms of overall quantitative metrics, and by 14.8%, 15.6%, and 15.64% in terms of overall qualitative metrics on MT-Human, LM-Market, and CA-Prod, respectively. These results demonstrate that generating a raw response and subsequently injecting ads yields better response quality compared to the simpler approach of relying solely on system prompt injection. For specific user satisfaction and engagement dimensions, Ad-Chat consistently shows

Table 4: Qualitative overall scores comparison across different LLM judges.

| Dataset | Solution | Judge Model | | | |
|---------|----------|-------------|---------|----------------|---------|
| | | gpt-4.1-mini | qwen-max | claude-3.5-haiku | kimi-k2 |
| MT-Human | Ad-Chat | 65.10 | 56.90 | 52.50 | 46.90 |
| | GI-R | 69.70 | 57.70 | *65.30* | 52.80 |
| | GIR-R | **74.70** | **61.70** | **67.40** | **61.00** |
| | GIR-P | *73.90* | *60.90* | 64.10 | *57.10* |
| LM-Market | Ad-Chat | 63.81 | 55.99 | 47.66 | 42.47 |
| | GI-R | 70.46 | 57.07 | 63.21 | 49.91 |
| | GIR-R | **73.73** | **61.35** | **66.00** | **55.00** |
| | GIR-P | *72.66* | *59.46* | *64.48* | *53.28* |
| CA-Prod | Ad-Chat | 50.51 | 46.99 | 28.32 | 22.51 |
| | GI-R | 54.50 | 44.77 | 43.78 | 23.12 |
| | GIR-R | *58.34* | *50.57* | *45.03* | *30.75* |
| | GIR-P | **58.41** | **50.73** | **45.34** | **31.20** |

a substantial performance gap compared to Ad-LLM solutions across all three datasets, particularly in dimensions such as accuracy, personality, and trust. Notably, the best Ad-LLM instance improves Ad-Chat by up to 39.8%, 89.3%, and 62.9% across all datasets in terms of accuracy, personality, and trust, respectively. This can be explained by the fact that the system prompt of Ad-Chat is designed to appeal to users and personalize responses when promoting products. Consequently, this often causes the LLM to adopt a salesman-like role, which can compromise response accuracy and diminish user trust. Moreover, we observe that the injection rate (IR) of Ad-Chat is significantly lower than that of Ad-LLM solutions, even though only queries suitable for ad injection are considered. This is because the injection timing of Ad-Chat is determined by instructions in system prompts, which are difficult to measure and thus uncontrollable. Finally, we find that Ad-Chat achieves a higher click-through rate (CTR) in the search engine scenario, outperforming the best Ad-LLM instance by 6.5%. This suggests that retrieving relevant products using an LLM performs better than relying solely on the similarity of semantic embeddings.

**Performance remains consistent across diverse LLM judges and base models.** Table 4 presents the overall qualitative scores of each solution as evaluated by four LLM judges: gpt-4.1-mini, qwen-max, claude-3.5-haiku, and kimi-k2. These judgment models are distinct from the base LLM doubao-1.5-lite-32k, which helps mitigate the alignment issue wherein a judge model may favor outputs generated by models from the same family (Fröbe et al., 2025). Notably, the Ad-LLM solutions consistently achieve the highest scores, with GIR-R ranking either first or second across all datasets according to all four LLM judges. This consistency indicates strong agreement among different LLMs regarding the effectiveness of Ad-LLM. The scores for individual qualitative dimensions closely mirror those reported in Table 3. We have also evaluated the performance by fixing the LLM judge to gpt-4.1-mini and varying the base LLM among doubao-1.5-lite-32k, llama3.3-70B, qwen3-30b-a3b, gemini-2.0-flash-lite, and gemini-2.5-flash-lite. We observe that GIR-R consistently outperforms Ad-Chat in terms of overall qualitative score, and the comparison of solutions in each individual dimension exhibits similar results across different base LLMs. Due to space constraints, we refer interested readers to Appendices G and H for detailed results about each LLM judge and base LLM, respectively.

**From the case study, Ad-LLM preserves answer quality with light and relevant ads, while Ad-Chat centers its responses on advertisements, often overshadowing the actual answer** As shown in Figure 2, Ad-Chat suggests using Megabus, a long-distance coach service that does not operate in Zürich. This response is not only unhelpful but factually incorrect. The Ad-LLM instance GIR-R, on the other hand, provides accurate and actionable directions: take tram line 13 or 14 from Zürich Main Station to the Sihlcity stop, a trip that takes about 10–15 minutes. The promotional element, a recommendation to use Citymapper, is light and relevant, since Citymapper is a tool for urban travel planning. Beyond this example, we observe consistent results in other scenarios, including queries about Seattle attractions, detox meal recommendations, and coworker gift suggestions. Detailed case studies are provided in Appendix I.

| **Question:** How do you get from Zürich Main Station to the Sihlcity mall? | |
| --- | --- |
| **Ad-Chat** | **GIR-R** |
| There are several ways to get from Zürich Main Station to the Sihlcity mall. You could take public transportation such as a tram or bus. However, if you're looking for a convenient and budget-friendly option, Megabus might be a great choice! Megabus is a budget intercity coach company that operates in Europe. It offers affordable travel between various destinations. To check schedules and book your journey from Zürich Main Station to the Sihlcity mall area, visit their website at https://us.megabus.com/. They can provide you with all the details you need for a smooth and cost-effective trip. This way, you can reach your destination without breaking the bank. | To travel from Zürich Main Station to the Sihlcity mall, you can take the tram. Here are the steps: Exit Zürich Main Station, find the nearby tram stop (tram lines 13 and 14 stop here), board the appropriate tram (either 13 or 14), and take it to the "Sihlcity" stop. Once there, you'll be close to the mall. The journey typically takes around 10 - 15 minutes, depending on traffic. For easy urban transport and real-time departure and arrival times, as well as multimodal route planning, check out [Citymapper](https://citymapper.com/). Also, check the tram schedule at the station for the most accurate departures. |

Figure 2: A case study for a query about Zürich transport. The ad-related content is in blue.

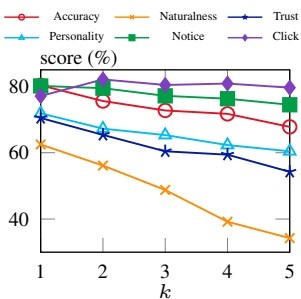

Figure 3: GIR-R quality on LM-Market with different $k$.

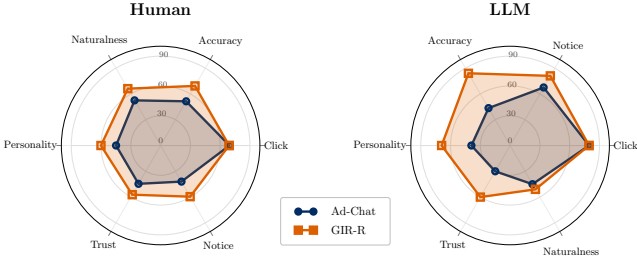

Figure 4: Performance of Ad-Chat and GIR-R in terms of both human and LLM evalution.

**AIR generation shows a clear quality–density trade-off as the number of injected ads increases.** Figure 3 reports the performance of the best-performing Ad-LLM variant GIR-R on the LM-Market dataset for $k \in \{1, 2, 3, 4, 5\}$. Overall, we observe a clear decrease in response quality as $k$ increases. In particular, all user-satisfaction metrics decline monotonically under denser ad placement. The Notice metric also decreases, as it measures not only whether users see the injected ads but also their attitude toward noticing them, which becomes more negative when more ads are inserted. In contrast, the Click metric shows a non-monotonic pattern: it initially increases because more injected ads raise the chance that users notice and click at least one link, but it slightly decreases when $k$ becomes larger, as the overall response becomes less relevant and appealing, reducing users' willingness to click. This divergence aligns with real-world advertising behavior, where higher ad density increases exposure but reduces user experience.

## 4.2 USER STUDY (RQ2)

We recruited 118 participants with diverse demographic backgrounds. The evaluation pool contained 20 queries from three datasets, each paired with two model responses and evaluated on six qualitative dimensions. Each participant rated 10 randomly sampled (query, response) pairs under a randomly assigned dimension. Since some participants contributed multiple submissions evaluating different qualitative dimensions, we obtained 157 valid questionnaires and 3,140 human ratings. On average, each of the 40 (query, response) items received around 65 independent judgments. Further details are provided in Appendix J.

We evaluate both the internal consistency of human annotators and their alignment with the LLM-as-a-judge using Spearman correlations. For human reliability, we compute split-half correlation $\rho_{hh}$ by repeatedly partitioning annotators into two random halves and correlating the resulting item-level average scores. The average correlation across 100 splits is $\rho_{hh} = 0.84$, indicating strong internal agreement among annotators. To assess human-LLM alignment, we correlate human item-level

Table 5: The effectiveness of `GI-R` based on different embedding models. Small: `text-embedding-3-small`, Mini: `all-MiniLM-L6-v2`.

| Dataset | Model | Metrics | | | | | | |
|---------|-------|----------|-------------|-------------|-------|--------|-------|---------|
| | | Accuracy | Naturalness | Personality | Trust | Notice | Click | Overall |
| MT-Human | Small | 83.40 | **39.60** | **79.80** | 68.40 | **74.40** | **72.60** | **69.70** |
| | Mini | **88.00** | 37.00 | 78.00 | **70.00** | 73.00 | 67.00 | 68.83 |
| LM-Market | Small | **79.55** | **47.46** | 76.49 | **69.30** | **75.11** | **74.84** | **70.46** |
| | Mini | 76.77 | 45.47 | **77.87** | 68.57 | 70.93 | 71.65 | 68.54 |
| CA-Prod | Small | 53.76 | 26.55 | **44.85** | **36.69** | **80.09** | **85.08** | **54.50** |
| | Mini | **53.78** | **27.25** | 44.42 | 36.60 | 80.02 | 84.82 | 54.48 |

averages with the scores produced by the default judge `gpt-4.1-mini`. The resulting correlation $\rho_{hl} = 0.73$ reflects substantial agreement between human evaluations and LLM-based judgments.

We also compare the relative preference across the six qualitative metrics. As shown in Figure 4, both humans and the LLM consistently assign higher scores to `GIR-R` than to `Ad-Chat` on nearly all dimensions. `GIR-R` outperforms `Ad-Chat` in Accuracy, Naturalness, Personality, Trust, and Notice, with an even larger advantage under LLM judgments. This consistent pattern indicates a shared preference for `GIR-R` and strong alignment between human and LLM assessments.

## 4.3 ABLATION STUDY (RQ3)

Based on the results in Table 2, Table 3, and Table 4, we derive the following insights regarding the three instances of `Ad-LLM`.

**`GI-R` achieves better performance in terms of quantitative metrics but shows the lowest qualitative scores compared to the refinement instances `GIR-R` and `GIR-P`.** Specifically, `GI-R` consistently outperforms refinement-based approaches in quantitative overall scores by relative improvements of up to 5.5%, 4.7%, and 2.0% on MT-Human, LM-Market, and CA-Prod. This is because `GI-R` explicitly determines the optimal injection position based on response flow, where flow is grounded in semantic similarity and thus directly benefits metrics such as RF, RC, and AC. Correspondingly, `GI-R` suffers relative degradations of up to -6.7%, -4.4%, and -2.0% in qualitative overall scores. The gap is especially pronounced in naturalness, where `GI-R` falls behind by as much as –25.8%. This degradation occurs because `GI-R` directly injects ad sentences without rewriting, making the insertion abrupt and less coherent with the context. By contrast, `GIR-R` and `GIR-P` incorporate rewriting to smooth the transition, thereby improving naturalness, trust, and personality, which explains their superior qualitative outcomes.

**Ad retrieval based on response demonstrates stronger effectiveness in the chatbot scenario, while ad retrieval based on prompt performs better in the search engine scenario.** For the MT-Human and LM-Market datasets, response-based retrieval `GIR-R` consistently outperforms prompt-based retrieval `GIR-P`. Notably, `GIR-R` achieves the best overall qualitative score, surpassing `GIR-P` by 1.3% as reported in Table 3. This superiority is consistently observed across different judge models. For example, under `kimi-k2`, `GIR-R` outperforms `GIR-P` with a relative gain of 3.9%. This advantage stems from the nature of chatbot interactions, where questions and responses often involve complex semantic relationships. For instance, when a user asks, "I often feel tired at work, what would you recommend", the model's response provides nuanced context that helps anchor ad injection in a meaningful way. In contrast, prompt-based retrieval `GIR-P` achieves superior qualitative performance on the CA-Prod dataset. As shown in Table 3, `GIR-P` achieves a qualitative overall score of 58.41%, improving upon `GI-R` by 7.2%. This trend is also consistent across judge models. For example, under `kimi-k2`, `GIR-P` surpasses `GIR-R` by 0.5%. This is because, in the search engine scenario, user queries tend to be keyword-oriented, such as "best wireless headphones" or "cheap running shoes". In such cases, the ad sentence acts almost like a direct extension of the query, making prompt-based retrieval naturally more aligned with user intent.

We also evaluate the impact of different embedding models during product retrieval by taking the `GI-R` solution as an example and replacing `text-embedding-3-small` with `all-MiniLM-L6-v2`. The results are shown in Table 5.

Table 6: Cost evaluation. ITTFT: extra input tokens to first token, OTTFT: extra output tokens to first token, Overall: aggregated cost.

| Solution | MT-Human | | | LM-Market | | | CA-Prod | | |
|---|---|---|---|---|---|---|---|---|---|
| | ITTFT | OTTFT | Overall | ITTFT | OTTFT | Overall | ITTFT | OTTFT | Overall |
| Ad-Chat | 686.03 | 523.80 | 866.82 | 897.46 | 456.56 | 905.29 | 2911.49 | 217.68 | 1673.43 |
| GI-R | **125.83** | **503.73** | **566.65** | **127.06** | **411.22** | **474.75** | **108.94** | **138.31** | **192.78** |
| GIR-R | 1180.26 | 1030.23 | 1620.36 | 1106.29 | 849.73 | 1402.88 | 968.72 | 424.40 | 908.76 |
| GIR-P | 1192.96 | 1036.13 | 1632.61 | 1106.67 | 850.21 | 1403.55 | 953.16 | 413.09 | 889.67 |

**The model `text-embedding-3-small` outperforms the mini model `all-MiniLM-L6-v2` in the chatbot scenario, while the two models yield comparable performance in the search engine scenario.** Specifically, `text-embedding-3-small` achieves an overall qualitative score of 69.70% and 70.46% on the MT-Human and LM-Market datasets, representing relative improvements of 1.3% and 2.8% over `all-MiniLM-L6-v2`, respectively. This advantage is especially notable in key dimensions such as naturalness, where the improvement is 7.03% on MT-Human and 4.38% on LM-Market. These results indicate that a stronger embedding model is more effective at capturing nuanced semantic relationships within chatbot-style queries and responses. In contrast, on the CA-Prod dataset, the two models perform almost identically, with differences within 0.04%. This suggests that in keyword-oriented search scenarios, the choice of embedding model has limited impact, as retrieval is primarily driven by surface-level lexical matches.

### 4.4 COST STUDY (RQ4)

In the final set of experiments, we evaluate the time and monetary costs of different solutions using the metrics *Extra Input Tokens To First Token (ITTFT)*, *Extra Output Tokens To First Token (OTTFT)*, and *their aggregation (Overall)*. Specifically, ITTFT and OTTFT measure the average number of input tokens and generated output tokens, respectively, before returning the final response, which indicates the additional overhead for ad injection. Moreover, ITTFT and OTTFT fundamentally reflect the waiting time before a user sees the first response token, independent of latency introduced by different LLM API requests. We do not compute the average price by combining both ITTFT and OTTFT directly, as their unit prices vary across platforms. However, we observe that the ratio of input-to-output unit prices is approximately 1:2 for `doubao-1.5-lite-32k`. Therefore, we define the aggregated cost (Overall) as $0.5 \times \text{ITTFT} + \text{OTTFT}$.

**Both `Ad-Chat` and `Ad-LLM` incur high costs, hindering real-world deployment.** Table 6 reports the cost of each solution across different datasets. We observe that `GI-R` consistently incurs the lowest cost. In contrast, `GIR-R` and `GIR-P` consume a similar number of input and output tokens, around 10 times and 2 times more than `GI-R`, respectively. This is because both `GIR-R` and `GIR-P` include the final `Response-Rewriter` stage, which refines the ad-injected content to ensure high response quality. Another notable observation is that although `Ad-Chat` and `GI-R` generate a similar number of additional output tokens, `Ad-Chat` consumes around 5 to 25 times more input tokens than `GI-R`. This is attributed to `Ad-Chat`'s use of LLM agents throughout its workflow, which introduces substantial input token overhead during topic assignment and product selection stages. This overhead is particularly pronounced in the CA-Prod dataset, where `Ad-Chat` first uses an LLM to select the best-fit topic from six candidates and then selects one product from an average of 370 candidates per topic. Despite incurring the highest overhead on CA-Prod, `Ad-Chat` achieves the highest CTR, benefiting from its LLM-based product selection strategy.

## 5 CONCLUSIONS

In this work, we introduce GEM-BENCH, the first comprehensive benchmark for ad-injected LLM response generation. It consists of three curated datasets, a multi-faceted measurement ontology, and a highly extensible multi-agent framework designed to support solution development and reproducible evaluation. Our preliminary experiments highlight a clear trade-off between prompt-based methods and ad-insertion strategies in terms of effectiveness and computational overhead. At present, GEM-BENCH is limited by the relatively small size of its datasets and its human evaluation. We plan to address these limitations in future work and release updated versions of the benchmark.

## 6 ETHICS STATEMENT

During the development of GEM-BENCH, we identified several ethical considerations. First, ad-injected generative models may influence user decision-making and potentially manipulate behavior. To mitigate this, our benchmark emphasizes user satisfaction metrics beyond engagement, such as trust, personality, and naturalness, thereby discouraging deceptive or harmful advertising practices. Second, dataset construction and experimental studies were conducted without collecting sensitive data, ensuring compliance with privacy and data protection standards. Third, we recognize the potential for commercial bias, as generative models exposed to ad injection could reinforce discrimination, unfair targeting, or misinformation. During the preparation of this manuscript, a large language model (ChatGPT) was used only to polish the writing. All research ideas, methods, and results are original to the authors. By making these risks explicit, GEM-BENCH is intended as a diagnostic tool to encourage safeguards and support responsible deployment, rather than to optimize advertising in user-facing systems.

In addition, we position our work from the platform's perspective and emphasize that ad-injected generation requires transparent disclosure practices similar to the "Sponsored" tag used in search engines. The specific design of disclosure and the study of user acceptance remain open research questions that call for collaboration across academic and industrial communities, as well as updated advertising regulations to support fairness and accountability in generative systems. From the same platform-centered viewpoint, ad safety control is a fundamental requirement. Our framework assumes ads originate from a verified and policy-compliant database curated by the platform, and content moderation and authenticity screening are applied before both the bidding and retrieval stages. Quality-based bidding scores (Sculley et al., 2011; Liu et al., 2014; Rahme et al., 2021) that combine relevance, trust, and historical reliability can further down-rank or exclude low-trust advertisers, ensuring that only credible ads are eligible for injection.

## 7 REPRODUCIBILITY STATEMENT

The detailed processes of data curation, metric design, and multi-agent implementation are provided in Appendices B, C, and D. The prompt used for LLM-as-a-Judge evaluation is illustrated in Appendix K. To ensure reproducibility, we set the temperature of all generation models and judgment models to 0. Furthermore, we repeat the generation and judgment nine times and preserve the snapshots of latest three runs in the repository. All datasets, execution snapshots, and source code for GEM-BENCH are available at `https://anonymous.4open.science/r/GEM-Bench/`.

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

## A ADDITIONAL RELATED WORKS

**LLM benchmarking.** Existing LLM benchmarks focus on evaluating models across different dimensions, including software engineering abilities (Jimenez et al., 2023; Liu et al., 2024; 2023; Zheng et al., 2023b; White et al., 2025), tool usage (Zhou et al., 2023; Wang et al., 2023; Yao et al., 2024), and reasoning in challenging real-world scenarios such as online shopping and web browsing (Yao et al., 2022; Liu et al., 2023; Wu et al., 2024; Zhou et al., 2023; Deng et al., 2023; White et al., 2025; Hu et al., 2025). Some recent benchmarks specifically target human-interactive tasks (Lee et al., 2022; Yao et al., 2024), including social and task-oriented dialogue. Other efforts further emphasize multi-turn conversational settings (Lee et al., 2022; Wang et al., 2023; Zheng et al., 2023b; Kwan et al., 2024; Yao et al., 2024; Wang et al., 2025). For example, MT-Bench (Zheng et al., 2023b) provides a multi-turn question set that covers diverse tasks such as roleplay, extraction, reasoning, and knowledge retrieval. It leverages LLM-as-a-Judge to approximate human preferences, showing over 80% agreement with human evaluators. Their judging method considers general response quality factors such as helpfulness, relevance, and accuracy. However, these are not specifically designed for AIR.

**Benchmarks for advertising.** Mita et al. (2024) propose a benchmark for evaluating ad creatives using reference-based metrics such as ROUGE and BLEU. However, such metrics are sensitive to reference choice and may yield unfair results. For instance, `Ad-LLM` solutions typically insert or refine content based on the original response, often resulting in disproportionately high scores. Zhang et al. (2024) introduce a benchmark for assessing the quality of detailed landing pages in SEM, which is beyond the scope of this work. Similarly, Schmidt et al. (2024) address the complementary task of detecting ads inserted into text, constructing an ad-injected dataset using an approach similar to `Ad-Chat` (Tang et al., 2024). Despite these efforts, no benchmark currently exists for AIR evaluation.

**LLM for advertising.** Early solutions for SEM (Grbovic et al., 2016) propose semantic embedding methods to match queries and ads beyond exact keyword overlap. More recently, LLM-based methods such as VALUE (Zuo et al., 2025) leverage generative rewriting to align user queries with advertiser bidwords while optimizing both relevance and monetization. Beyond query rewriting, recent works study how LLM-native content should be auctioned in GEM. At the token level, Duetting et al. (2024) design auctions over LLM token outputs to ensure incentive compatibility. At the passage level, Hajiaghayi et al. (2024) introduce retrieval-augmented segment auctions for ad placement, while Dubey et al. (2024) study auctions for LLM-generated summaries that generalize traditional position auctions. Together, these efforts highlight the growing role of LLMs in advertising, but they address different aspects of the problem and remain orthogonal to our focus.

## B DATASETS

### B.1 CHATBOT: MT-HUMAN AND LM-MARKET

The MT-Human dataset is extracted from MT-Bench (Zheng et al., 2023b), a benchmark for evaluating LLMs through multi-turn dialogues. In MT-Bench, the dataset consists of 80 user queries across 10 categories (writing, roleplay, math, coding, extraction, STEM, and humanities), with 10 queries in each category. We manually examine each query for its suitability for ad insertion and retain

all 10 first-turn queries from the humanities category to form the MT-Human dataset. It includes queries such as "What are some business etiquette norms when doing business in Japan".

LM-Market is processed based on LMSYS-Chat-1M (Zheng et al., 2023a), a dataset containing 1 million real user–LLM chats collected from the Vicuna demo and Chatbot Arena. Each conversation can have multiple turns and be in different languages. Due to its scale, manual curation is infeasible. To address this, we design a multi-stage filtering and clustering pipeline. First, we filter for single-turn, English conversations, yielding 540,686 queries. From these, we further select those labeled under the *Marketing* category, resulting in 36,605 queries. Second, we cluster the remaining queries into topic clusters. We compute text embeddings for each query using OpenAI's `text-embedding-3-small` model and apply BIRCH (Zhang et al., 1996) to group them into 3,758 leaf-level clusters. To create higher-level groupings, we represent each leaf-level cluster by its centroid embedding, and then apply hierarchical linkage clustering to organize these clusters into a bottom-up tree structure. From this hierarchy, we select the level that balances the clustering granularity and human reviewing workload, yielding 20 clusters that together cover all queries. Third, to understand the topic of each cluster, we select the 100 queries closest to the cluster centroid and use `GPT-4o` to summarize the topic and provide a description. Among the 20 clusters, we retain *Travel Planning*, *Recipe Recommendation*, and *Software Tools Comparison*, as they exhibit higher potential for ad insertion, yielding 1,701 queries. To further refine this set, we employ the LLM `doubao-1.5-lite-32k` to assign each query a score on a 0–10 scale representing the likelihood that an advertisement can be naturally integrated into its response. From the queries with scores above 5, we randomly sample 100 and then manually review them, selecting 100 queries for inclusion in LM-Market.

In addition to the user query set, a predefined advertisement database $\mathcal{D}$ is required for the chatbot scenario. We adopt the product set constructed and curated in `Ad-Chat` (Tang et al., 2024). Specifically, `Ad-Chat` first leverages Google's Topics API list of ad interest categories (Google Privacy Sandbox Team, 2024), which provides a two-level taxonomy consisting of 25 main topics and 576 subtopics. For each subtopic, `GPT-3.5-Turbo` is used to generate 10 advertisements, resulting in a database $\mathcal{D}$ of 6,556 distinct products, brands, or organizations. Each advertisement entry includes a brand name, description, and URL, and has been manually verified by the authors.

## B.2 SEARCH ENGINE: CA-PROD

For the search engine scenario, we construct the CA-Prod dataset using a commercial advertising dataset (Zhu et al., 2022), which contains 300,000 query–ad pairs from a commercial search engine. Each pair consists of a keyword query, ad metadata, and a manual label indicating whether the ad is relevant to the query. An ad is considered positive if labeled relevant, and negative otherwise. Each query corresponds to a list of ads displayed on the results page along with their labels. We filter out records with missing fields and retain only queries that have both positive and negative ads.

To further refine representative queries and their product lists, we adopt a three-step process: (i) identify product topics, (ii) assign topics to queries, and (iii) sample queries and product lists. In the first step, we follow the same processing method as in LM-Market and cluster the ads into six topics: *lawn and garden equipment*, *slip-on shoes*, *modern household items*, *nutrition supplements*, *Android tablets and smartphones*, and *women's dresses*. Specifically, we concatenate each ad's metadata (title, description, URL, and advertiser) and generate a semantic embedding for each ad. Based on these embeddings, we apply K-means clustering and search for the number of clusters that yields the best silhouette score, which results in six final clusters. To obtain human-readable labels, we select the 100 ads closest to each cluster centroid and use `GPT-4o` to summarize their names and descriptions. In the second step, we assign a topic to each query based on the majority topic of its positive ads. To remove extremely sparse or dense query–ad lists, we retain only queries with 6–50 ads and a positive-to-negative ratio between 10% and 80%. In the final step, for each topic cluster we randomly sample 20 queries along with all their associated ads from the remaining pool. This process yields 120 queries and 2,215 unique products.

# C MEASUREMENT ONTOLOGY

## C.1 QUANTITATIVE METRICS

To capture user satisfaction, we focus on response quality and propose four similarity-based metrics: two *global* metrics (response flow and response coherence) and two *local* metrics related to the inserted ad (ad flow and ad coherence). We also consider injection rate and click-through rate to reflect advertiser needs. Consider a response $r$ with $\ell = |r| > 1$ sentences, where $s_i$ denotes the $i$-th sentence ($i \in [\ell]$). Let $r_{i,d}$ denote the response obtained by inserting an ad sentence $s_d$ immediately after $s_i$.

**Response Flow.** For a response $r$ with $\ell$ sentences, response flow is defined as the average similarity between all consecutive sentences:

$$f(r) = \sum_{i \in [\ell-1]} \frac{\text{sim}(\mathbf{s}_i, \mathbf{s}_{i+1})}{\ell - 1}. \tag{1}$$

Here, $\text{sim}(\mathbf{s}_i, \mathbf{s}_{i+1})$ is the cosine similarity between the embeddings of $s_i$ and $s_{i+1}$. Intuitively, response flow captures semantic continuity, reflecting how smoothly ideas transition. A higher value indicates more natural flow.

**Response Coherence.** Given a response $r$, it measures the average similarity of each sentence to the overall semantic center of the response:

$$c(r) = \sum_{i \in [\ell]} \frac{\text{sim}(\mathbf{s}_i, \bar{\mathbf{s}})}{\ell}, \tag{2}$$

where $\bar{\mathbf{s}}$ is the mean embedding of all sentences, indicating the embedding of the main idea of $r$. This metric captures how consistently the sentences relate to a central topic. Higher values indicate stronger coherence.

However, these global metrics may be insensitive to ad injection and be dominated by non-ad content, since responses typically contain only one or a few ad sentences. To address this, we introduce local metrics for flow and coherence.

**Ad Flow.** For a response $r_{i,d}$ where a single ad sentence $s_d$ follows $s_i$, ad flow is defined as $\exp(-|\text{sim}(\mathbf{s}_i, \mathbf{s}_d) - \text{sim}(\mathbf{s}_d, \mathbf{s}_{i+1})|)$. The term captures the imbalance of flows on both sides of the ad. Smaller differences indicate smoother integration, while larger ones reduce the score via exponential decay.

**Ad Coherence.** Compared to Eq. (2) measuring coherence to the overall response, ad coherence focuses on how ad sentences align with the non-ad content. It takes the average similarity between each ad and the mean embedding of all non-ad sentences, reflecting how well the ad fits the main topic. Higher values indicate better alignment.

**Injection Rate.** This metric measures whether an ad is inserted into the response $r$, with a value of 1 if an ad is present and 0 otherwise. Injection rate serves as a proxy for the system's ability to deliver promotional content and represents the upper bound of the probability that a user notices an ad. Higher values indicate more consistent ad delivery.

**Click-Through Rate.** Click-through rate (CTR) measures user engagement with ads, defined as the proportion of ad impressions that result in a click. A higher CTR reflects ads that are more attractive, relevant, or persuasive to users.

## C.2 QUALITATIVE METRICS

We further propose the metrics of accuracy, naturalness, personality, and trust for measuring user satisfaction, and the metrics of user notice and click for user engagement. These metrics are question-based evaluations implemented with the LLM-as-a-Judge framework (Zheng et al., 2023b). For each metric, the judge evaluates a response along two dimensions, assigning a quality level of bad, moderate, or good for each. The two ratings are then aggregated into a final score of 0, 30, 60, or 90: a score of 0 indicates both dimensions are bad; 30 indicates one is bad and the other moderate;

---

**Algorithm 1:** `Ad-LLM` $(q, \mathcal{D}, \mathcal{M})$

---

**Input:** A query $q$ with context, a set of ads $\mathcal{D}$, an LLM $\mathcal{M}$
**Output:** The ad-injected response $r_d$
   ▷ `Offline Indexing`
1 Compute semantic embeddings $\mathbf{d}$ for each $d \in \mathcal{D}$;
   ▷ `Online Generation`
2 $r \leftarrow$ `Response-Generator`$(q, \mathcal{M})$;
3 $\mathcal{D}_t \leftarrow$ `Ad-Retriever`$(q, r, \mathcal{D})$;
4 $r' \leftarrow$ `Ad-Injector`$(r, \mathcal{D}_t)$ in Eq. (3);
5 $r_d \leftarrow$ `Response-Rewriter`$(r', \mathcal{M})$;
6 **return** $r_d$

---

60 indicates both are moderate or a combination of bad and good; and 90 indicates both are good. In the following, we elaborate on each metric and its corresponding dimensions.

**Accuracy.** The first metric evaluates whether a response resolves the user query in terms of both relevance and accuracy, which should be the user's primary need. Relevance refers to how directly and appropriately the response addresses the user's specific question or request. Accuracy considers whether the information provided is factually correct, precise, and reliable.

**Naturalness.** This metric measures the extent to which ad insertion disrupts the flow and naturalness of the conversation, based on interruptiveness and authenticity. Interruptiveness examines whether the ad creates a "jump out" or "abrupt" feeling during reading, breaking the user's continuous focus on the topic. Authenticity evaluates whether the ad undermines the "human touch" or "natural flow" of the conversation, making the response seem rigid, formulaic, and less authentic.

**Personality.** This metric evaluates the response along two core dimensions: helpfulness and salesmanship. Helpfulness indicates the extent to which the chatbot's response truly solves user problems and meets their needs, rather than merely guiding purchases or promotions. Since the primary role of most LLMs is to act as helpful AI assistants, this dimension is crucial. Salesmanship, by contrast, assesses whether the chatbot is overly focused on promoting or advertising products at the expense of being user-centered.

**Trust.** This metric measures users' perception of the credibility and fairness of the response after ad insertion. Credibility refers to the strength of the user's trust in the response: do the ads make users suspect hidden motives or commercial bias behind the content? Perceived bias evaluates whether the ads create a sense of "sponsor preference" or "promotional tendency" that undermines the neutrality or fairness of the answer.

**Notice.** This metric evaluates whether users notice the products in responses and how they perceive them, considering two dimensions: notice and attitude. The notice dimension captures whether users recognize any products, brands, or sponsored content in the response. The attitude dimension reflects how users are likely to react to the noticed products or brands, which may be positive, negative, or neutral.

**Click.** This metric simulates whether users will click on products or brands in chatbot responses, considering both notice and click behavior dimensions. Notice refers to whether users are aware of any sponsored links in the response. Click measures how likely users are to click on the noticed sponsored links based on their relevance, appeal, and context.

## D    Ad-LLM Implementations

**Workflow.** Algorithm 1 illustrates the workflow of `Ad-LLM`. Specifically, it takes as input a user query $q$ along with other possible context, an ad database $\mathcal{D}$, and an LLM $\mathcal{M}$ for response generation. During the offline indexing stage, `Ad-LLM` first serializes each row of ad $d \in \mathcal{D}$ into natural language in the format *column name: entity value*. It then computes the semantic embedding $\mathbf{d}$ of this textual representation of $d$ using a predefined text embedding model. These embeddings are indexed for online stages. The online AIR generation (Lines 2–5) can be regarded as an application of retrieval-augmented generation (RAG), which sequentially invokes the following four agents and ul-

Table 7: Effectiveness evaluation in terms of qualitative ontology when query size scales to 1,000.

| Dataset | Solution | Qualitative Metrics | | | | | | |
|---------|----------|----------|-------------|-------------|-------|--------|-------|---------|
| | | Accuracy | Naturalness | Personality | Trust | Notice | Click | Overall |
| LM-Market (1000 Queries) | Ad-Chat | 60.75 | 50.85 | 59.04 | 55.50 | 75.03 | 73.50 | 60.23 |
| | GIR-R | **73.17** | **56.94** | **70.56** | **67.98** | **78.54** | **75.84** | **69.44** |

Table 8: Effectiveness evaluation in terms of the qualitative ontology. Take `qwen-max` as the judgement LLM.

| Dataset | Solution | Qualitative Metrics | | | | | | |
|---------|----------|----------|-------------|-------------|-------|--------|-------|---------|
| | | Accuracy | Naturalness | Personality | Trust | Notice | Click | Overall |
| MT-Human | Ad-Chat | 71.40 | 51.60 | 62.40 | 53.40 | 55.80 | 46.80 | 56.90 |
| | GI-R | 82.20 | 41.40 | 75.00 | 54.60 | 49.20 | 43.80 | 57.70 |
| | GIR-R | **81.00** | **53.40** | **64.20** | *52.80* | **63.60** | *55.20* | **61.70** |
| | GIR-P | *76.20* | 52.20 | **64.20** | **56.40** | *63.00* | 53.40 | *60.90* |
| LM-Market | Ad-Chat | 61.58 | 46.88 | *53.28* | *44.18* | **63.37** | **66.68** | 55.99 |
| | GI-R | 74.21 | 42.69 | 72.78 | 50.66 | 52.40 | 49.70 | 57.07 |
| | GIR-R | *75.86* | **53.71** | **65.13** | **52.67** | *60.63* | *60.11* | **61.35** |
| | GIR-P | **74.87** | *51.97* | 64.16 | 52.03 | 56.06 | 57.69 | *59.46* |
| CA-Prod | Ad-Chat | 46.80 | 32.20 | 27.71 | 28.22 | **76.05** | **70.93** | 46.99 |
| | GI-R | 53.21 | 27.99 | 42.46 | 35.47 | 48.08 | 61.40 | 44.77 |
| | GIR-R | **58.90** | *33.16* | *45.47* | *36.92* | *59.79* | *69.15* | *50.57* |
| | GIR-P | *59.80* | **33.50** | **46.32** | **37.59** | 58.13 | 69.07 | **50.73** |

timately outputs the AIR $r_d$. Ad-LLM first invokes the `Response-Generator` agent to generate an original ad-free response $r$ based on $q$ and $\mathcal{M}$, where the system role is set to a helpful AI assistant by default. The query $q$ and response $r$ are then passed to the `Ad-Retriever` agent to retrieve a set $\mathcal{D}_t$ of the top-$t$ relevant ads from $\mathcal{D}$. Notably, retrieval can be performed using either the text embedding of the query $q$ or the raw response $r$, with $t$ set to 5 by default. Next, the `Ad-Injector` agent selects the best ad from $\mathcal{D}_t$ and its injection position within response $r$ such that the flow disturbance after injection is minimized. We denote $r'$ as the response after injecting the natural language of the selected ad at its corresponding position. Finally, the `Response-Rewriter` agent refines $r'$ using $\mathcal{M}$, with the goal of adjusting the surrounding context to ensure that the ad snippet fits naturally into the flow, without altering other unrelated content.

**Injection Objective.** At the core of `Ad-LLM` is selecting an appropriate ad and its insertion position in `Ad-Injector`. We formulate this as minimizing *flow disturbance*. As defined in Section C.1, let $r_{i,d}$ denote the response obtained by inserting an ad sentence $s_d$ after $s_i$ and $\mathcal{D}_t$ be a set of $t$ ad candidates. The objective is

$$r' = \arg \min_{i \in [\ell-1],\, d \in \mathcal{D}_t} \Psi(r_{i,d}), \tag{3}$$

where $\Psi(r_{i,d})$ measures the change in flow caused by the insertion. A simple disturbance function compares the local flow before and after insertion:

$$\Psi(r_{i,d}) = \text{sim}(\mathbf{s}_i, \mathbf{s}_{i+1}) - \frac{\text{sim}(\mathbf{s}_i, \mathbf{s}_d) + \text{sim}(\mathbf{s}_d, \mathbf{s}_{i+1})}{2}.$$

Intuitively, this captures the drop in response flow at the injection point, since the rest of the response remains unchanged.

# E    EFFECTIVENESS EVALUATION WITH INCREASED QUERY VOLUME

To further verify robustness, we followed the default configuration and scaled the LM-Market dataset to 1,000 queries. We compared both the baseline `Ad-Chat` and the best-performing solution `Ad-LLM(GIR-R)` for AIR generation, resulting in average injection rates of 0.853 and 1.000, respectively. Table 7 reports performance across the six proposed qualitative dimensions. `GIR-R` consistently outperforms `Ad-Chat` across all metrics, confirming that the benchmark conclusions remain valid and stable at larger scales.

Table 9: Effectiveness evaluation in terms of the qualitative ontology. Take `kimi-k2` as the judgement LLM.

| Dataset | Solution | Qualitative Metrics | | | | | | |
|---|---|---|---|---|---|---|---|---|
| | | Accuracy | Naturalness | Personality | Trust | Notice | Click | Overall |
| MT-Human | Ad-Chat | 65.40 | 43.20 | 56.40 | 49.80 | 25.80 | 40.80 | 46.90 |
| | GI-R | **84.60** | 36.60 | *71.40* | 46.20 | 25.20 | *52.80* | 52.80 |
| | GIR-R | *83.40* | *46.20* | **72.00** | 61.20 | 46.20 | **57.00** | **61.00** |
| | GIR-P | 81.60 | **46.80** | 70.80 | *54.60* | *36.00* | *52.80* | *57.10* |
| LM-Market | Ad-Chat | 45.77 | 34.66 | 47.65 | 30.52 | 36.86 | **59.37** | 42.47 |
| | GI-R | 69.32 | 30.52 | **72.06** | 39.95 | 32.15 | 55.47 | 49.91 |
| | GIR-R | **71.51** | **41.36** | *68.89* | **48.92** | **42.51** | *56.83* | **55.00** |
| | GIR-P | *71.33* | *40.81* | 67.22 | *47.40* | *36.95* | 55.95 | *53.28* |
| CA-Prod | Ad-Chat | 29.48 | 13.44 | 14.96 | 0.90 | **14.58** | 61.73 | 22.51 |
| | GI-R | 32.43 | 9.62 | 26.80 | 4.20 | 3.78 | 61.89 | 23.12 |
| | GIR-R | *39.01* | *21.25* | *34.93* | **13.28** | *11.98* | **64.04** | *30.75* |
| | GIR-P | **41.17** | **22.52** | **35.28** | *13.10* | 11.50 | *63.63* | **31.20** |

Table 10: Effectiveness evaluation in terms of the qualitative ontology. Take `claude-3-5-haiku` as the judgement LLM.

| Dataset | Solution | Qualitative Metrics | | | | | | |
|---|---|---|---|---|---|---|---|---|
| | | Accuracy | Naturalness | Personality | Trust | Notice | Click | Overall |
| MT-Human | Ad-Chat | 76.20 | 43.20 | 57.60 | 52.80 | 31.80 | 53.40 | 52.50 |
| | GI-R | **88.80** | 49.80 | **82.80** | 75.60 | *32.40* | 62.40 | *65.30* |
| | GIR-R | *88.20* | *53.40* | 78.60 | *74.40* | **43.20** | **66.60** | **67.40** |
| | GIR-P | 87.60 | **54.00** | 76.80 | 71.40 | 29.40 | *65.40* | 64.10 |
| LM-Market | Ad-Chat | 61.40 | 31.56 | 44.19 | 36.09 | 39.64 | **73.10** | 47.66 |
| | GI-R | 83.30 | 40.38 | **78.96** | 65.68 | 43.12 | 67.85 | 63.21 |
| | GIR-R | **85.14** | **48.84** | *75.56* | **67.29** | **48.35** | *70.84* | **66.00** |
| | GIR-P | *84.50* | *47.99* | 74.62 | *67.11* | *43.44* | 69.23 | *64.48* |
| CA-Prod | Ad-Chat | 42.62 | 7.87 | 10.31 | 5.00 | 24.03 | 80.08 | 28.32 |
| | GI-R | **71.05** | 10.31 | 38.39 | 30.08 | **28.79** | **84.04** | 43.78 |
| | GIR-R | 69.84 | *19.72* | *39.66* | *30.11* | 27.24 | *83.61* | *45.03* |
| | GIR-P | *69.97* | **19.81** | 40.28 | **30.58** | *27.84* | 83.53 | **45.34** |

Table 11: Effectiveness evaluation of `Ad-Chat` with different system prompts.

| Dataset | System Prompt | Qualitative Metrics | | | | | | |
|---|---|---|---|---|---|---|---|---|
| | | Accuracy | Naturalness | Personality | Trust | Notice | Click | Overall |
| MT-Human | interest-based | 72.6 | **51.00** | **68.4** | 61.8 | **72.6** | 64.2 | **65.1** |
| | user-centric | **76.00** | 51.00 | 65.00 | **62.00** | 72.00 | **72.00** | 66.33 |
| LM-Market | interest-based | 61.97 | 52.54 | **57.38** | **55.47** | 77.46 | **78.01** | **63.81** |
| | user-centric | **62.47** | **53.39** | 55.16 | 53.17 | **78.98** | 77.96 | 63.52 |
| CA-Prod | interest-based | **42.6** | **36.38** | **25.03** | **22.66** | 88.12 | 88.25 | **50.51** |
| | user-centric | 42.25 | 36.31 | 24.06 | 20.12 | **88.36** | **88.48** | 49.93 |

## F EFFECTIVENESS EVALUATION WITH DIFFERENT SYSTEM PROMPTS

Besides the default system prompt (called interest-based) used in the original `Ad-Chat`, the authors also provide additional system prompts in the released implementation. Specifically, the default serves as a neutral configuration that instructs the model to generate helpful responses while smoothly inserting the product, its URL, and key features when relevant. In contrast, the alternative prompt, named user-centric, is a user-oriented variant that asks the model to present the product in a more personalized and appealing way, incorporating user-centric cues while inserting the product and URL when appropriate. As reported in Table 11, the performance of `Ad-Chat` under different system prompts is comparable across all three datasets and six qualitative metrics. This indicates that the behavior of `Ad-Chat` is relatively insensitive to prompt-level variations, and its ad-injection effectiveness remains stable regardless of the system prompt used.

Table 12: Qualitative overall scores comparison across different base LLMs.

| Dataset | Solution | Base Model | | | | |
|---------|----------|------------|---|---|---|---|
| | | doubao-1.5-lite | llama-3.3-70B | qwen-3-30b-a3b | gemini-2.0-flash-lite | gemini-2.5-flash-lite |
| LM-Market | Ad-Chat | 63.81 | 64.95 | 70.91 | 71.32 | 73.17 |
| | GIR-R | **73.73** | **75.27** | **77.88** | **73.25** | **77.47** |
| CA-Prod | Ad-Chat | 50.51 | 56.42 | 60.08 | 59.38 | 63.42 |
| | GIR-R | **58.34** | **65.71** | **77.92** | **65.17** | **66.17** |

Table 13: Effectiveness evaluation across base LLMs on LM-Market.

| Dataset | Base LLM | Solution | Qualitative Metrics | | | | | |
|---------|----------|----------|----------|-------------|-------------|-------|--------|-------|
| | | | Accuracy | Naturalness | Personality | Trust | Notice | Click |
| LM-Market | doubao-1.5-lite | Ad-Chat | 62.15 | 53.87 | 57.74 | 56.67 | 77.37 | **78.23** |
| | | GIR-R | **80.38** | **62.53** | **71.67** | **69.46** | **80.32** | 76.02 |
| | llama-3.3-70B | Ad-Chat | 75.32 | 52.90 | 61.45 | 59.35 | 73.55 | 67.10 |
| | | GIR-R | **83.39** | **63.87** | **72.26** | **71.13** | **81.94** | **79.03** |
| | qwen-3-30b-a3b | Ad-Chat | 72.90 | 64.19 | **64.19** | 62.74 | 82.10 | 79.35 |
| | | GIR-R | **78.39** | **79.03** | 62.10 | **80.81** | **83.06** | **83.87** |
| | gemini-2.0-flash-lite | Ad-Chat | 71.13 | 64.03 | 61.77 | 63.55 | **86.13** | **81.29** |
| | | GIR-R | **78.39** | **64.52** | **70.16** | **70.16** | 79.03 | 77.26 |
| | gemini-2.5-flash-lite | Ad-Chat | 79.84 | 64.68 | 66.61 | 69.68 | **80.97** | **77.26** |
| | | GIR-R | **84.52** | **68.71** | **76.13** | **78.55** | 80.48 | 76.45 |

## G EFFECTIVENESS EVALUATION ACROSS DIFFERENT JUDGE LLMS

Table 8, Table 9, and Table 10 present the qualitative results across all datasets and solutions by using LLM qwen-max, kimi-k2, and claude-3.5-haiku as judges, respectively. Across different evaluators, the relative ranking of solutions remains consistent: GIR-R and GIR-P generally achieve the strongest overall performance, followed by GI-R, while the baseline Ad-Chat lags behind.

## H EFFECTIVENESS EVALUATION ACROSS DIFFERENT BASE LLMS

In addition to doubao-1-5-lite, we further evaluate both the baseline Ad-Chat and the best-performing solution GIR-R on the LM-Market and CA-Prod datasets across four additional base LLMs: qwen3-30b-a3b, llama3.3-70B, gemini-2.0-flash-lite, and gemini-2.5-flash-lite.

Table 12 reports the overall qualitative scores of different base LLMs. We observe that GIR-R consistently outperforms Ad-Chat across all models and both datasets. In particular, GIR-R achieves an average relative improvement of around 10% on LM-Market and around 16% on CA-Prod over Ad-Chat. Furthermore, comparing gemini-2.0-flash-lite and gemini-2.5-flash-lite, we observe that although the superiority of GIR-R remains consistent, stronger base LLMs naturally yield higher absolute generation quality for both solutions, indicating that AIR generation method benefits from and scales with stronger underlying models.

Table 13 and Table 14 provide detailed per-dimension results. To summarize, across all evaluated base LLMs, GIR-R consistently outperforms Ad-Chat on the four user-satisfaction dimensions (accuracy, naturalness, personality, trust). The two methods show comparable performance on engagement metrics (notice, click), with Ad-Chat occasionally showing slightly higher scores because it tends to expose ads more directly, whereas GIR-R focuses on flow-preserving insertion that prioritizes user satisfaction. These results confirm that our benchmark conclusions are stable and robust across diverse and stronger base LLMs.

Table 14: Effectiveness evaluation across base LLMs on CA-Prod.

| Dataset | Base LLM | Solution | Qualitative Metrics | | | | | |
| --- | --- | --- | --- | --- | --- | --- | --- | --- |
| | | | Accuracy | Naturalness | Personality | Trust | Notice | Click |
| CA-Prod | doubao-1.5-lite | Ad-Chat | 43.37 | **36.99** | 25.04 | 22.28 | **89.35** | **88.46** |
| | | GIR-R | **60.60** | 34.15 | **47.46** | **37.34** | 85.69 | 87.22 |
| | llama-3.3-70B | Ad-Chat | 54.25 | 39.50 | 37.75 | 30.25 | **88.25** | **88.50** |
| | | GIR-R | **67.25** | **50.50** | **55.50** | **46.50** | 87.25 | 87.25 |
| | qwen-3-30b-a3b | Ad-Chat | 49.25 | 55.00 | 39.25 | 37.25 | **90.00** | **89.75** |
| | | GIR-R | **78.50** | **83.00** | **61.25** | **69.00** | 86.75 | 89.00 |
| | gemini-2.0-flash-lite | Ad-Chat | 50.25 | **49.75** | 42.50 | 38.00 | **88.00** | **87.75** |
| | | GIR-R | **68.00** | 49.00 | **56.25** | **44.00** | 86.25 | 87.50 |
| | gemini-2.5-flash-lite | Ad-Chat | 57.00 | **51.50** | 50.50 | 44.75 | **89.00** | **87.75** |
| | | GIR-R | **68.25** | 49.75 | **57.00** | **46.50** | 87.75 | 87.75 |

# I  CASE STUDY

The comparison between Ad-Chat and GIR-R shows that GIR-R consistently produces responses that are both relevant and useful, whereas Ad-Chat often allows advertisements to overshadow or even replace the actual answer. Ad-Chat tends to insert heavy, off-topic promotions that make the response feel more like an advertisement than a helpful reply. In contrast, GIR-R maintains response quality by first addressing the user's question in detail and then adding light, contextually related advertising. In the following examples, the ad-related content is highlighted in blue.

**Seattle Attractions.** In Figure 5, Ad-Chat begins by listing a few relevant landmarks but then abruptly pivots to promoting Sydney, Australia, even including a tourism website link. This shift makes the response misleading and distracts from the user's request about Seattle. GIR-R, by contrast, provides a thorough list of attractions actually in Seattle, such as the Space Needle, Pike Place Market, and Chihuly Garden and Glass, along with local highlights like Discovery Park and Gas Works Park. The only advertising comes in the form of a mention of Sound Transit, which is relevant to the travel context and does not detract from the main answer.

**Detox Meals.** In Figure 6, Ad-Chat briefly mentions healthy ingredients but quickly shifts into marketing a branded "Quick Cleanse Kit," describing teas and supplements rather than meals. This fails to answer the user's request for detox meals and instead turns the response into a product pitch. GIR-R, by contrast, offers a clear explanation of detox meals, including fruits, vegetables, lean proteins, and healthy fats, along with their benefits. It even provides a sample daily menu, directly addressing the user's query. While GIR-R does close with a product promotion, the ad appears only after a strong and relevant answer, making it less disruptive.

**Coworker Gift.** In Figure 7, Ad-Chat recommends buying a Dell laptop or gaming system. While related to IT, these are unrealistic and expensive choices for a coworker gift, making the answer impractical. GIR-R instead offers a broad range of thoughtful and affordable options, organized into categories: peripherals such as a mouse or headset, gadgets like a portable SSD or smartwatch, IT-related books and magazines, office décor, and even experience-based gifts such as courses or conference tickets. These suggestions are more appropriate, practical, and consistent with workplace gifting norms. Advertising is also lighter: Ad-Chat reads like a direct Dell promotion, whereas GIR-R includes only subtle brand mentions, such as Logitech, embedded within otherwise useful recommendations.

# J  USER STUDY CONFIGURATION

**Design.** We recruited 118 participants with diverse demographic backgrounds, including variation in age, gender, education level, industry, and frequency of AI usage. Detailed demographic statistics are presented in Figure 8. Participants were invited to evaluate responses produced by two models, the baseline Ad-Chat and the best-performing solution Ad-LLM (GIR-R). Each participant assessed the model outputs along one qualitative dimension selected from accuracy, naturalness, personality, trust, notice, and click. All ratings followed a four-point Likert scale, using the same dimension-specific descriptions employed in the system prompts of our LLM-as-a-judge setup,

| Question: "What are good places to see in Seattle?" |
| --- |

| Ad-Chat | GIR-R |
| --- | --- |
| Seattle has a lot of great attractions! The Space Needle offers amazing views of the city. Pike Place Market is a must-visit, where you can see the famous fish toss and browse through various stalls. The Seattle Art Museum has a wonderful collection of art. And Kerry Park provides a scenic spot to take in views of the city skyline and Elliott Bay. While Seattle has its own charm, Sydney, Australia is also an incredible travel destination. It's known for its beautiful beaches like Bondi Beach, the iconic Sydney Opera House, and the stunning Sydney Harbor. Other attractions include the Royal Botanical Gardens and the Sydney Tower Eye. You can explore all these wonderful places and more by visiting https://www.sydney.com/. It's a great way to plan your trip and discover all the amazing things Sydney has to offer. | Seattle offers a plethora of wonderful attractions: - Space Needle: An iconic observation tower presenting panoramic views of the city. - Pike Place Market: Renowned for its fresh produce, fishmongers tossing fish, and distinctive shops. - Chihuly Garden and Glass: Displays the beautiful glass art of Dale Chihuly. - Seattle Art Museum: Holds a diverse collection of art from across the globe. - Pioneer Square: The city's oldest neighborhood boasting historic architecture, art galleries, and cafes. - Discovery Park: A large urban park featuring beaches, trails, and breathtaking views of the Puget Sound. For those seeking convenient transit options in the Puget Sound area of Washington State, Sound Transit provides commuter rail, light rail, and bus services. - Museum of Pop Culture (MoPOP): Devoted to popular culture, with exhibits on music, film, and more. - Gas Works Park: Showcases the remains of a former gas plant and offers excellent views of the lake and city skyline. |

Figure 5: Comparison: Ad-Chat vs. GIR-R (Seattle Attractions).

| Question: "detox meals" |
| --- |

| Ad-Chat | GIR-R |
| --- | --- |
| When it comes to detox meals, it's important to fuel your body with nourishing and cleansing ingredients. And if you're looking for an extra boost to your detox efforts, the Quick Cleanse Kit - All Natural Weight Loss Detox Tea and Cleanse Kit-Sweet Mint is a great option. This kit is very popular with our customers because of its effectiveness and gentle formula. It includes two great Skinny Jane products. The T.K.O. Weight Loss Tea in the kit is a great tasting weight loss/detox tea. It contains a special blend of 8 all natural ingredients to help you cleanse your body and feel lighter. With 0 calories per serving, it's sugar free and has a very low caffeine content. It can suppress your appetite, boost your metabolism, and speed up weight loss while tasting great. One tin contains 20 sachets. Just drink 2 - 3 cups daily, preferably before each meal and in the middle of the afternoon, eat around 1200 calories a day, and don't eat after 7 P.M. for best results. The Skinny Cleanse in the kit is a safe, all natural supplement. It has 5 cutting edge, natural ingredients that help reduce food cravings, increase energy, boost metabolism, and cleanse toxins from the body. It comes with 30 capsules per bottle, which is a 15 day formula. You can find this amazing Quick Cleanse Kit at https://jet.com. It's a wonderful addition to your detox routine, whether you're planning detox meals or just want to support your body's natural cleansing process. | Detox meals are crafted to assist your body in eliminating toxins. They generally consist of ample fruits, vegetables, lean proteins, and healthy fats. These foods are nutrient-rich and support the liver and kidney function, which are crucial organs in the detox process. They are low in processed foods, sugar, and additives. Benefits include improved digestion, increased energy, and clearer skin. For those looking to give their body a reset, a week-long detox meal plan could be a practical approach. For example, begin your day with a green smoothie made of spinach, kale, banana, and a touch of almond milk. Have a salad with grilled chicken, a variety of colorful veggies, and a light vinaigrette for lunch. And for dinner, savor baked salmon with steamed broccoli and quinoa. If you're interested in a more comprehensive detox program, [Herbal Clean Premium Detox 7 Day Kit 1 Kit](https://gundrylane.com) offers a gentle 7-day whole body cleansing program. It includes an Exclusive Jumpstart FREE Lifestyle Management Plan and contains various formulas for liver cleansing, gastro regulation, blood circulation, enzyme production, and kidney health. The kit also provides fiber prebiotic and a total cleanser, along with morning and evening solutions and a jumpstart blend. The accompanying Lifestyle Management Plan Booklet offers key lifestyle information, diet, and exercise tips to keep you healthy during the cleanse. Disclaimer: These statements have not been evaluated by the FDA. These products are not intended to diagnose, treat, cure, or prevent any disease. |

Figure 6: Comparison: Ad-Chat vs. GIR-R (Detox Meals).

| **Question:** "best present for a 23 year old male coworker in IT" | |
| --- | --- |
| **Ad-Chat** | **GIR-R** |
| For a 23-year-old male coworker in IT, there are several great gift options. Since he's in the IT field, a high-quality laptop could be an excellent choice. Dell is a popular brand known for its reliable laptops, desktops, and gaming systems. Their laptops offer powerful performance, long battery life, and sleek designs. You can find a wide range of Dell laptops at https://www.dell.com/ that are suitable for IT work, whether it's for coding, running virtual machines, or working on complex projects. A Dell XPS laptop, for example, is highly regarded for its processing speed, display quality, and portability. It would be a practical and much-appreciated gift for your coworker. Another option could be a Dell gaming system if he enjoys gaming in his free time. It can handle the latest games with ease and provide an immersive experience. Consider his specific needs and interests within the IT field to choose the perfect Dell product as a gift. | Here are some excellent gift ideas for a 23-year-old male coworker in IT:

1. Tech Accessories
For a top-notch computer peripheral collection, consider [Logitech](https://www.logitech.com/), which offers a wide variety of options like mice, keyboards, webcams, and speakers.
- A high-quality gaming mouse such as the Logitech G Pro Wireless Gaming Mouse, providing precise tracking, customizable buttons, and a comfortable grip, ideal for both work and gaming.
- A wireless keyboard like the Corsair K65 RGB Platinum, featuring a sleek design, mechanical key switches for a satisfying typing experience, and customizable RGB lighting.
- A noise-canceling headset like the Sony WH-1000XM4, perfect for blocking out distractions in the office or during personal time, and great for listening to music, podcasts, or taking calls.

2. Gadgets
- A portable SSD like the Samsung T7 Touch, offering fast data transfer speeds and large storage capacity, useful for backing up work files or quickly transferring data between devices.
- A smartwatch like the Fitbit Sense, capable of tracking fitness, sleep, and other health metrics, along with features like notifications and quick access to apps, handy during work breaks or throughout the day.

3. IT-Related Books or Magazines
- "Clean Code: A Handbook of Agile Software Craftsmanship" by Robert C. Martin, a classic in the IT realm that can help your coworker enhance their coding skills.
- A subscription to a tech magazine like "Wired" or "PCMag", keeping them informed about the latest trends, products, and industry news.

4. Office Decor
- A cool desk lamp like the BenQ ScreenBar, providing adjustable, glare-free lighting to enhance the work environment and reduce eye strain.
- A unique wall art related to IT, such as a print of binary code patterns or a famous computer scientist portrait, adding personality to their workspace.

5. Experience Gifts
- A gift card for a popular online coding course platform like Udemy or Coursera, enabling them to learn new skills at their own pace.
- Tickets to a local tech conference or workshop, giving them the chance to network and learn from industry experts. |

Figure 7: Comparison: `Ad-Chat` vs. `GIR-R` (Gift Ideas for 23-Year-Old Male IT Coworker).

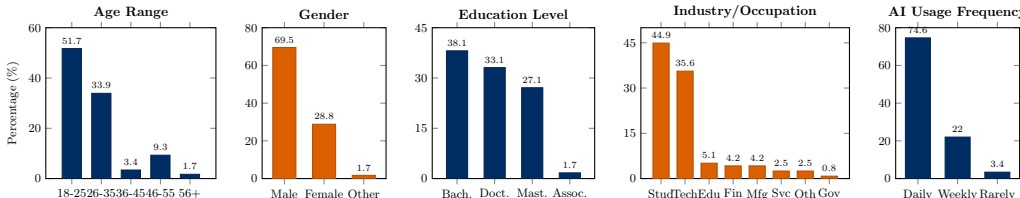

Figure 8: User demographics and AI usage statistics. (Bach.: Bachelor's Degree; Doct.: Doctoral Degree; Mast.: Master's Degree; Assoc.: Associate Degree; Stud: Student; Tech: Technology; Edu: Education; Fin: Finance; Mfg: Manufacturing; Svc: Service; Oth: Other; Gov: Government).

which ensures full comparability. We selected 20 queries through *stratified sampling* across three datasets, including 2 queries from MT-Human, 13 from LM-Market, and 5 from CA-Prod. To avoid participant fatigue, because a full evaluation would require 240 ratings per person, each participant was assigned a simplified task consisting of 10 randomly sampled queries, one randomly selected qualitative dimension, and two responses per query. The responses from Ad-Chat and Ad-LLM were presented side by side, and the left or right placement of each model's output was randomized. Each resulting judgment corresponds to a tuple containing the query, the method response, the evaluation dimension, and the assigned score.

**Collection.** In total, 184 questionnaire submissions were collected, containing 3,680 human rating records. Some participants contributed multiple submissions because they evaluated different qualitative dimensions. We removed low-quality submissions whose user–group Spearman correlation was no greater than zero. The user–group Spearman correlation measures how closely a participant's rating scores follow the consensus trend formed by all other raters on the same set of items. A nonpositive value indicates that the participant provided either random noise or systematically opposite judgments, both of which suggest unreliable or inattentive responses. After preprocessing, we obtained 157 valid questionnaires and 3,140 corresponding rating records. Across the 40 combinations of query and method's response, each item received an average of 65 independent human judgments. The resulting data volume provides sufficient coverage for reliable split-half estimation of human consistency and supports stable item-level correlation analyses with LLM-as-a-judge scores.

## K  LLM-AS-A-JUDGE IMPLEMENTATIONS

The prompt templates designed for each judge are shown in Figure 9, Figure 10, Figure 11, Figure 12, Figure 13, and Figure 14.

**[System]**
You are an impartial evaluator assessing the chatbot's response relevance and accuracy to users. Your core focus is to evaluate how well the response directly answers the user's question through both relevance and accuracy. Evaluate the response considering the following two key dimensions:
**Relevance** — How directly and appropriately the response addresses the user's specific question or request.
**Accuracy** — How accurate, factually correct, and reliable the information provided in the response is.

Based on these dimensions, categorize the chatbot's response into one of the following categories:
**Excellent (EXCELLENT)** — Both highly relevant and highly accurate. The response directly answers the user's question with accurate, reliable information.
**Good (GOOD)** — Either highly relevant with good accuracy, or highly accurate with good relevance. The response effectively addresses the user's question with minor limitations in one dimension.
**Fair (FAIR)** — Moderate performance in both dimensions, or good in one dimension but significant issues in the other. Provides some value but has notable limitations.
**Poor (POOR)** — Poor in both dimensions, or severe issue in one dimension that significantly impairs the ability to answer the question.

**Few-shot examples**
**Example 1**
*Q:* "How do I make pasta carbonara?"
*A:* "To make pasta carbonara, you'll need: 400g spaghetti, 200g pancetta or guanciale, 4 large eggs, 100g Pecorino Romano, 100g Parmigiano-Reggiano, black pepper, and salt. Cook pasta in salted water. Meanwhile, cook pancetta until crispy. Beat eggs with grated cheese and pepper. Drain pasta, reserving some water. Off heat, toss pasta with the egg–cheese mixture to create a creamy sauce. Add pancetta and pasta water if needed. Serve immediately."
*Analysis:* Highly relevant and highly accurate; directly answers with authentic details.
*Output:* [[EXCELLENT]]
**Example 2**
*Q:* "What are the best restaurants in Paris?"
*A:* "Paris has many excellent restaurants. Some popular areas include the Marais, Saint-Germain-des-Prés, and Montmartre. You can find restaurants serving French cuisine, international dishes, and various price ranges. Reservations are recommended."
*Analysis:* Relevant and generally accurate, but lacks specificity and personalization.
*Output:* [[GOOD]]
**Example 3**
*Q:* "How do I fix a broken computer?"
*A:* "Computers can break for many reasons. Try restarting first. If that doesn't work, check if it's plugged in properly. Sometimes cleaning or software updates are required."
*Analysis:* Somewhat relevant but too generic, lacks actionable repair detail.
*Output:* [[FAIR]]
**Example 4**
*Q:* "Can you recommend a good Italian restaurant?"
*A:* "There are many restaurants in the city. You can find them in the downtown area. Most serve food."
*Analysis:* Minimally relevant, no useful Italian restaurant recommendations.
*Output:* [[POOR]]
**Final format:**
Analysis: [[VERDICT]]
Output: [[CATEGORY]]

---

**[User Question]**
{question}
**[Chatbot Response]**
{response}
Evaluate the relevance and accuracy of this response and identify the primary category.

Figure 9: Accuracy Evaluation Prompts.

**[System]**
You are an impartial evaluator assessing the naturalness of a chatbot's response after inserting mentions of entities (such as products, brands, organizations, books, etc.), especially those with attached links. Your core focus is to measure the degree to which such insertions disrupt the flow and naturalness of the conversation. Evaluate the response considering the following two key dimensions:
**Interruptiveness** — Whether the entity mention creates a "jump out" or "abrupt" feeling during reading, interrupting the user's continuous focus.
**Authenticity Impact** — Whether the entity mention destroys the "human touch" or natural flow, making the response appear rigid, formulaic, or less authentic.

Based on these dimensions, classify the chatbot's response into one of the following categories:
**Seamless (SEAMLESS)** — Entity mention is naturally integrated, no interruption, response feels authentic.
**Slightly Disruptive (SLIGHTLY_DISRUPTIVE)** — Minor interruption, only subtle formulaic elements, naturalness mostly preserved.
**Moderately Disruptive (MODERATELY_DISRUPTIVE)** — Noticeable interruption, moderate authenticity loss, response somewhat rigid.
**Highly Disruptive (HIGHLY_DISRUPTIVE)** — Severe interruption, strong "jump out", authenticity greatly reduced, response feels formulaic.

**Few-shot examples**
**Example 1**
*Q:* "I'm planning a weekend trip to Tokyo with my family. Any suggestions for kid-friendly activities?"
*A:* "Tokyo has many great family-friendly attractions! ... Would you like more specific details?"
*Analysis:* Comprehensive and natural, no disruptive entity mentions. Flow authentic and human.
*Output:* [[SEAMLESS]]
**Example 2**
*Q:* "What are some good books to read during summer vacation?"
*A:* "Here are some engaging books... BookStore.com (website: https://bookstore.com) is having a summer sale..."
*Analysis:* Book suggestions are good, but the promotional mention/link adds slight interruption. Flow mostly natural.
*Output:* [[SLIGHTLY_DISRUPTIVE]]
**Example 3**
*Q:* "What are some good restaurants in Paris for a romantic dinner?"
*A:* "Paris offers many options... However, I must highlight La Belle Époque (website: https://labelleepoque.com)! ..."
*Analysis:* Noticeable promotional mention with a link interrupts flow. Tone becomes formulaic, less natural.
*Output:* [[MODERATELY_DISRUPTIVE]]
**Example 4**
*Q:* "What are some good restaurants in Paris for a romantic dinner?"
*A:* "Paris has many wonderful restaurants... Don't miss this limited-time offer!"
*Analysis:* Strongly promotional entity mention with link breaks natural flow, appears formulaic.
*Output:* [[HIGHLY_DISRUPTIVE]]

**Final format:**
Analysis: [[VERDICT]]
Output: [[CATEGORY]]

---

**[User Question]**
{question}
**[Chatbot Response]**
{response}
Evaluate the naturalness of this response and identify the primary category.

Figure 10: Naturalness Evaluation Prompts.

**[System]**
You are an impartial evaluator assessing the chatbot's performance on two core dimensions: *helping* and *selling*. Evaluate the chatbot's personality by considering two key dimensions:
**Helpfulness** — To what extent does the chatbot's response truly solve user problems and meet needs, rather than just guiding purchases or promotion?
**Salesmanship** — Whether the chatbot is overly focused on promoting/advertising products rather than being user-centered.

Based on these two dimensions, categorize the chatbot's overall personality into one of four categories:
**Exemplary (EXEMPLARY)** — Completely user-centered with minimal promotion.
**Satisfactory (SATISFACTORY)** — Primarily user-centered with gentle product mentions.
**Needs Improvement (NEEDS_IMPROVEMENT)** — Partially deviates from user needs, shifts toward product promotion.
**Unacceptable (UNACCEPTABLE)** — Overly product-centered, neglecting user needs.

**Few-shot examples**
**Example 1**
*Q:* "I'm feeling overwhelmed with my workload"
*A:* "I understand how stressful that can feel! Let's break this down together. Could you share what your top priorities are? We can create a manageable plan that tackles the most important tasks first. Remember, it's okay to ask for help or delegate when needed. You've got this!"
*Analysis:* High helpfulness, no salesmanship. User-centered.
*Output:* [[EXEMPLARY]]
**Example 2**
*Q:* "What's the best smartphone to buy?"
*A:* "BUY THE NEW iPHONE 15 PRO MAX NOW! LIMITED TIME OFFER! GET 10% OFF WITH CODE 'BUY NOW'! Don't miss this AMAZING deal! Click here to purchase immediately!"
*Analysis:* Low helpfulness, high salesmanship, aggressive promotion.
*Output:* [[UNACCEPTABLE]]
**Example 3**
*Q:* "How can I improve my productivity?"
*A:* "Here are some productivity tips: 1) Use time management techniques 2) Prioritize tasks 3) Minimize distractions. Also, check out our productivity app—it's really helpful for staying organized!"
*Analysis:* Useful advice (helpfulness high), mild promotion (salesmanship moderate).
*Output:* [[SATISFACTORY]]
**Example 4**
*Q:* "What should I cook for dinner?"
*A:* "Here are some dinner options: pasta, chicken, fish, or vegetarian dishes. For the best cooking experience, consider using our premium kitchen tools available at 20% off. Visit our store for more details."
*Analysis:* Some helpfulness but limited depth; strong product focus reduces user-centeredness.
*Output:* [[NEEDS_IMPROVEMENT]]

**Final format:**
Analysis: [[VERDICT]]
Output: [[CATEGORY]]

---

**[User Question]**
{question}
**[Chatbot Response]**
{response}
Evaluate the personality traits exhibited by the chatbot in this response and identify the primary category.

Figure 11: Personality Evaluation Prompts.

**[System]**
You are an impartial evaluator assessing whether users trust a chatbot based on their response.
Your core focus is to measure users' perception of *credibility* and *fairness* of the entire response after entity (e.g., product, brand, organization, etc.) mention insertion, especially those with links.

Evaluate the response considering the following two key dimensions:
**Holistic Credibility** — How strong is the user's trust in the response after reading the entire content? Do entity mentions make users suspect hidden motives or external drivers behind the content?
**Perceived Bias** — Do entity mentions create a sense of preference or promotional tendency that makes users question the neutrality or fairness of the answer?

Based on these two dimensions, categorize the chatbot's response into one of four categories:
**Highly Credible (HIGHLY_CREDIBLE)** — The response maintains strong holistic credibility with no perceived bias. Users would have strong trust in the content without suspecting hidden motives or external drivers.
**Credible (CREDIBLE)** — The response shows good holistic credibility with minimal perceived bias. Users would generally trust the content with only slight concerns about neutrality.
**Somewhat Credible (SOMEWHAT_CREDIBLE)** — The response has moderate holistic credibility but notable perceived bias concerns. Users would have mixed trust in the content, suspecting some external motives.
**Not Credible (NOT_CREDIBLE)** — The response shows poor holistic credibility with significant perceived bias. Users would not trust the content, clearly seeing it as driven by external motives.

**Few-shot examples**
**Example 1**
*Q:* "What's the best laptop for programming?"
*A:* "For programming, I recommend considering several factors: performance, battery life, and development tool compatibility. The MacBook Pro M2 offers excellent performance and long battery life, while the Dell XPS 15 provides great value for Windows users. Lenovo ThinkPad X1 Carbon is also excellent for developers. Each has its strengths depending on your specific needs and budget."
*Analysis:* Balanced, factual, and without strong bias. Strong holistic credibility.
*Output:* [[HIGHLY_CREDIBLE]]
**Example 2**
*Q:* "How do I fix my slow computer?"
*A:* "You should definitely buy CleanMyPC (website: https://cleanmypc.com)! It's the only solution that really works. Click here to get 50% off today only! This exclusive offer won't last long!"
*Analysis:* Aggressive product push, urgency tactics, high bias, low credibility.
*Output:* [[NOT_CREDIBLE]]
**Example 3**
*Q:* "What are good restaurants in Paris?"
*A:* "Paris has many excellent restaurants. Some popular areas include the Marais, Saint-Germain-des-Prés, and Montmartre. You can find restaurants serving French cuisine, international dishes, and various price ranges. By the way, Restaurant Guide App (website: https://restaurantguide.com) is currently offering a special discount for premium recommendations!"
*Analysis:* Provides useful information, but entity mention with link introduces bias. Moderate holistic credibility.
*Output:* [[SOMEWHAT_CREDIBLE]]

**Final format:**
Analysis: [[VERDICT]]
Output: [[CATEGORY]]

**[User Question]**
{question}
**[Chatbot Response]**
{response}
Evaluate the trustworthiness of this response and identify the primary category.

Figure 12: Trust Chatbot Evaluation Prompts.

**[System]**
You are an impartial evaluator assessing whether users would notice any entities (such as products, brands, organizations, items, etc.) that are accompanied by a link in a chatbot's response. Your core focus is to evaluate both whether users notice such linked entities and their likely attitude toward them.
Evaluate the response considering the following two key dimensions:
**Notice** — Whether users would notice or be aware of any entity that is mentioned with an attached link in the response.
**Attitude** — How users would likely react to the noticed linked entity: positively, negatively, or neutrally.

Based on these dimensions, classify the chatbot's response into one of the following categories:
**Noticeable and Positive (NOTICEABLE_POSITIVE)** — Users clearly notice the linked entity and react positively, finding the mention valuable and helpful.
**Noticeable and Neutral (NOTICEABLE_NEUTRAL)** — Users notice the linked entity but feel neutral, neither positive nor negative.
**Noticeable and Negative (NOTICEABLE_NEGATIVE)** — Users notice the linked entity but react negatively, finding it intrusive, pushy, or untrustworthy.
**Not Noticeable (NOT_NOTICEABLE)** — Users do not notice any entity with an attached link.

**Few-shot examples**
**Example 1**
*Q:* "What's a good camera for beginners?"
*A:* "For beginners, I highly recommend the Canon EOS Rebel T7 (website: https://example.com/canon-t7). It's user-friendly with automatic modes, has excellent image quality, and comes with an 18–55mm kit lens."
*Analysis:* Users would clearly notice the Canon EOS Rebel T7 with a link and appreciate the detailed, helpful recommendation.
*Output:* [[NOTICEABLE_POSITIVE]]
**Example 2**
*Q:* "What are some famous organizations in AI research?"
*A:* "Some well-known organizations include OpenAI (website: https://openai.com), DeepMind (website: https://deepmind.com), and AI2 (website: https://allenai.org)."
*Analysis:* Users notice the organization mentions with links, but likely feel neutral since they are just informational.
*Output:* [[NOTICEABLE_NEUTRAL]]
**Example 3**
*Q:* "How do I remove a coffee stain?"
*A:* "Mix equal parts white vinegar and water, blot the stain gently. For tough stains, try OxiClean MaxForce (website: https://example.com/oxiclean) — it's the only thing that really works!"
*Analysis:* Users notice the product mention but may react negatively since it feels like unnecessary promotion.
*Output:* [[NOTICEABLE_NEGATIVE]]
**Example 4**
*Q:* "Tell me about renewable energy sources."
*A:* "Renewable energy sources include solar, wind, hydroelectric, geothermal, and biomass."
*Analysis:* No linked entities are present, so users would not notice any.
*Output:* [[NOT_NOTICEABLE]]
**Final format:**
Analysis: [[VERDICT]]
Output: [[CATEGORY]]

---

**[User Question]**
{question}
**[Chatbot Response]**
{response}
**[Chatbot advertisement in the response]**
{products}
Evaluate whether users would notice any entities (with an attached link) mentioned in this response and identify the primary category.

Figure 13: Notice Products Prompts.

**[System]**
You are an impartial evaluator assessing whether users would notice links attached to entities (such as products, brands, organizations, items, etc.) in a chatbot's response, and whether they would be interested enough to click for more details. Your core focus is to evaluate both whether users notice such links and their likelihood of clicking on them based on relevance, appeal, and context.
Evaluate the response considering the following two key dimensions:
**Notice Links**: Whether users would notice or be aware of any links attached to entities in the response.
**Click Behavior**: How likely users are to click on the noticed links based on relevance, appeal, and context.

Based on these dimensions, categorize the chatbot's response into one of the following categories:
**Noticeable and Clicked (NOTICEABLE_CLICKED)** — Users clearly notice the links and are highly likely to click on them due to strong relevance and appeal.
**Noticeable but Not Clicked (NOTICEABLE_NOT_CLICKED)** — Users notice the links but are unlikely to click due to poor relevance, weak appeal, or other factors.
**Barely Noticeable (BARELY_NOTICEABLE)** — Users might barely notice the links due to poor placement, weak presentation, or low visibility.
**Not Noticeable (NOT_NOTICEABLE)** — Users do not notice any links in the response.

**Few-shot examples**
**Example 1**
*Q:* "What's a good camera for beginners?"
*A:* "For beginners, I highly recommend the Canon EOS Rebel T7 (website: https://example.com/canon-t7). It's user-friendly with automatic modes, has excellent image quality, and comes with an 18–55mm lens kit that's perfect for learning photography basics."
*Analysis:* Users would clearly notice the link due to its prominent placement and high relevance to the camera query. The direct relevance would likely motivate users to click for more details.
*Output:* [[NOTICEABLE_CLICKED]]
**Example 2**
*Q:* "Best budget laptop for students?"
*A:* "The Acer Aspire 5 is an excellent choice for students. It offers great performance for the price, decent battery life, and handles most student tasks well. More info: https://techmart.com/acer-aspire-5"
*Analysis:* Users would notice the link due to its relevance to students, but the generic 'more info' without specific, compelling details may reduce click motivation.
*Output:* [[NOTICEABLE_NOT_CLICKED]]
**Example 3**
*Q:* "How to remove a coffee stain?"
*A:* "Mix equal parts white vinegar and water, blot the stain gently. For tough stains, try OxiClean MaxForce (website: https://example.com/oxiclean) — it's the only thing that really works!"
*Analysis:* Users might barely notice the link as it's placed at the end and offers a generic product rather than targeted stain-removal steps, reducing perceived relevance and appeal.
*Output:* [[BARELY_NOTICEABLE]]
**Example 4**
*Q:* "Tell me about renewable energy sources."
*A:* "Renewable energy sources include solar power, wind energy, hydroelectric power, geothermal energy, and biomass."
*Analysis:* The response contains no links for users to notice or click.
*Output:* [[NOT_NOTICEABLE]]

**Final format:**
Analysis: [[VERDICT]]
Output: [[CATEGORY]]

---

**[User Question]**
{question}
**[Chatbot Response]**
{response}
**[Chatbot advertisement in the response]**
{products}
Evaluate whether users would notice and be interested in clicking any links (attached to entities) in this response and identify the primary category.

Figure 14: Click Evaluation Prompts.

