# OpenReview forum: "GEM-Bench: A Benchmark for Ad-Injected Response Generation within Generative Engine Marketing"
_ICLR.cc/2026/Conference — Submitted to ICLR 2026_

### Official Review · Reviewer_hUJt · 2025-10-30

**Soundness:** 2
**Presentation:** 2
**Contribution:** 2
**Rating:** 4
**Confidence:** 4

**Summary:**

This paper introduces GEM-Bench, a benchmark for evaluating ad-injected response (AIR) generation in both chatbot and search settings. It provides three curated datasets and a set of evaluation metrics designed to capture multiple dimensions of user satisfaction and engagement. The authors establish baseline results using a modular multi-agent framework. Experimental findings indicate that prompt-based ad insertion can improve engagement but may compromise accuracy and trust, whereas generate–then–insert approaches yield higher response quality at the cost of increased computational overhead.

**Strengths:**

- S1. The benchmark covers both conversational and search-oriented use cases, enabling broader applicability across domains.

- S2. The proposed evaluation metrics are multifaceted and reflect several relevant user experience dimensions.

- S3. The multi-agent baseline framework is conceptually simple, easy to extend, and provides a reproducible starting point for future work.

- S4. The experimental study is reasonably comprehensive and clearly illustrates trade-offs between prompt-based and multi-stage insertion strategies.

**Weaknesses:**

- W1. The evaluation relies entirely on LLM-as-judge scoring without any human annotation or preference validation, making it unclear how well the measured metrics align with real user perception.

- W2. The methodological contribution is relatively limited; the baseline (i.e., Ad-Chat) is straightforward and not previously peer-reviewed, which reduces the novelty.

- W3. Engagement evaluation is based on synthetic CTR estimates rather than real user interaction data, so the results reflect perceived clickability rather than actual conversion behavior.

**Questions:**

- Q1: For Figure 2, are the shown weaknesses of Ad-Chat due to inherent limitations of prompt-based insertion, or mainly due to the specific prompt used? Were alternative prompt designs evaluated?

---

> ### Author Response · Authors · 2025-11-20
> **Thanks for Reviewer hUJt’s comments**
>
> ### **Response to W1**
>
> Please see **Common Response #1**.
>
> ### **Response to W2**
>
> We thank the reviewer for raising this concern. We would like to clarify that the baseline **Ad-Chat is a peer-reviewed and accepted work**, published as *“Ads that Talk Back: Implications and Perceptions of Injecting Personalized Advertising into LLM Chatbots”* (IMWUT/Ubicomp 2025). We adopt it as the most relevant existing system for comparison, given its direct focus on ad-injected conversational agents. Regarding our methodological contribution, we kindly refer the reviewer to our detailed response to **Reviewer eZeW-W2**.
>
> ### **Response to W3**
>
> Please see **Common Response #2**.
>
> ### **Response to Q1**
>
> We thank the reviewer for the question. Besides the default system prompt (referred to as **interest-based**) used in the original Ad-Chat, the authors also provide additional system prompts in the released implementation. Specifically, the default prompt serves as a neutral configuration that instructs the model to generate helpful responses while smoothly inserting the product, its URL, and key features when relevant. In contrast, the alternative prompt, named **user-centric**, is a user-oriented variant that asks the model to present the product in a more personalized and appealing manner, incorporating user-centric cues while inserting the product and URL when appropriate. As reported in the table below, the performance of Ad-Chat under these different system prompts is comparable across all three datasets and all six qualitative metrics. **This suggests that the behavior of Ad-Chat is relatively insensitive to prompt-level variations.**
>
> | Dataset | System Prompt | Accuracy  | Naturalness  | Personality  | Trust  | Notice  | Click  | Overall |
> | --- | --- | --- | --- | --- | --- | --- | --- | --- |
> | MT-Human | interest-based | 72.60 | 51.00 | 68.40 | 61.80 | 72.60 | 64.20 | 65.10 |
> |  | user-centric | 76.00 | 51.00 | 65.00 | 62.00 | 72.00 | 72.00 | 66.33 |
> | LM-Market | interest-based | 61.97 | 52.54 | 57.38 | 55.47 | 77.46 | 78.01 | 63.81 |
> |  | user-centric | 62.47 | 53.39 | 55.16 | 53.17 | 78.98 | 77.96 | 63.52 |
> | CA_Prod | interest-based | 42.60 | 36.38 | 25.03 | 22.66 | 88.12 | 88.25 | 50.51 |
> |  | user-centric | 42.25 | 36.31 | 24.06 | 20.12 | 88.36 | 88.48 | 49.93 |

---

> ### Author Response · Authors · 2025-11-26
>
> Hi, thank you again for taking the time to review our submission. We were wondering if you might have any further comments or feedback. We truly appreciate your efforts and are happy to provide any additional information if needed. Thank you!

---

### Official Review · Reviewer_tDUn · 2025-10-31

**Soundness:** 2
**Presentation:** 3
**Contribution:** 3
**Rating:** 6
**Confidence:** 3

**Summary:**

The paper introduces GEM-BENCH, the first comprehensive benchmark framework for Ad-Injected Response (AIR) generation. GEM-BENCH constructs two datasets — MT-Human and LM-Market — for chatbot scenarios, as well as CA-Prod, which simulates the AI overview feature in search engines. The paper also proposes a multi-dimensional evaluation system that covers both user satisfaction and engagement. Experimental results demonstrate that multi-stage ad insertion (Generate–Inject–Rewrite) significantly improves user satisfaction, trust, and personality compared to prompt-based ad injection (Ad-Chat), although at a higher computational cost.

**Strengths:**

1.This paper is among the first to systematically define, benchmark, and evaluate the task of ad-injected response (AIR) generation for LLM-based systems. It addresses a highly relevant and timely challenge for the monetization of generative AI.
2.The proposed GEM-Bench is comprehensive and thoughtfully constructed, featuring curated datasets from multiple scenarios (chatbot and search) and a multi-faceted evaluation ontology that captures both user satisfaction and engagement.
3.The paper provides a systematic comparison between prompt-based and multi-step ad-injection methods. The analysis offers valuable insights into the inherent trade-offs, explaining the underlying reasons for the performance differences, such as the compromise of accuracy in prompt-based methods and the increased overhead in multi-step approaches.

**Weaknesses:**

1.The paper does not explicitly define the calculation methodology for key qualitative metrics (e.g., Accuracy, Naturalness) in the main text. While LLM-as-a-Judge is employed, the heavy reliance on this approach without a clear, transparent rubric in the primary description may raise concerns about the objectivity and reliability of the evaluations.
2.The benchmark relies exclusively on multiple LLMs as judges for subjective metrics like user satisfaction. The lack of human validation to corroborate these automated judgments is a significant limitation, as it risks propagating or amplifying biases inherent in the judge models themselves.
3.The chatbot datasets contain only 10 and 100 queries, respectively, which may limit statistical robustness and generalizability to diverse domains.

**Questions:**

1.How sensitive are the results to the number of ads (k > 1) inserted into each response?
2.Have you conducted any small-scale human evaluation to validate the consistency of LLM-as-a-Judge results?
3.How are factual claims within the injected advertisements themselves verified for correctness? While the paper evaluates factual consistency of the overall response, it remains unclear whether the factual validity of the advertisement content itself is separately verified or controlled.
4.Given the potential ethical concerns surrounding ad injection, are there reasonable mechanisms in place to filter out deceptive or fraudulent ads with misleading inducements?

---

> ### Author Response · Authors · 2025-11-20
> **Thanks for Reviewer tDUn's comments**
>
> ### **Response to W1**
>
> In our current design, each qualitative metric is implemented within the LLM-as-a-Judge framework and evaluated along **two well-defined dimensions**. Specifically, **Accuracy** evaluates whether the response directly answers the user’s question based on relevance and factual correctness. **Naturalness** measures how ad insertion affects conversational flow through interruptiveness and authenticity. **Personality** assesses whether the assistant remains user-centered by examining helpfulness and the extent of sales-driven behavior. **Trust** evaluates users’ confidence in the response by considering holistic credibility and perceived bias introduced by ads. **Notice** simulates whether users recognize the mentioned products using product awareness and their resulting attitude. **Click** estimates link-level engagement based on link awareness and the likelihood of clicking, given relevance and appeal. For each dimension, the judge assigns one of three quality levels: *bad*, *moderate*, or *good*. The two dimension-level judgments are then aggregated into a final discrete score using a unified rubric: **0** for both dimensions rated *bad*; **30** for one *bad* and one *moderate*; **60** for both *moderate* or a combination of *bad* and *good*; **90** for both *good*.
>
>
> ### **Response to W2**
>
> See **Common Response #1**.
>
> ### **Response to W3**
>
> We thank the reviewer for this valuable comment. While the released chatbot subsets contain 10 and 100 queries for reproducibility, our curated datasets actually include 1,701 queries for LM-Market. To further verify robustness, we followed the default configuration and scaled the LM-Market dataset to **1,000 queries**. We compared both the baseline Ad-Chat and the best-performing solution Ad-LLM (GIR-R) for ad-injected response generation. The table below reports performance across the six proposed LLM-judge dimensions. **Ad-LLM (GIR-R) consistently outperforms Ad-Chat across all metrics, confirming that the benchmark conclusions remain valid and stable at larger scales.**
>
> |  | Accuracy  | Naturalness  | Personality  | Trust  | Notice  | Click  | Overall |
> | --- | --- | --- | --- | --- | --- | --- | --- |
> | Ad-Chat | 60.75 | 50.85 | 59.04 | 55.50 | 75.03 | 73.50 | 60.23 |
> | GIR-R | **73.17** | **56.94** | **70.56** | **67.98** | **78.54** | **75.84** | **69.44** |
>
> ### **Response to Q1**
>
> We thank the reviewer for the constructive comment. We have included a sensitivity analysis to address this question.
>
> As described in Lines 867–870 of the submitted paper, we extend Ad-LLM to the **k-ad setting** via an iterative injection procedure. The table below reports the performance of the best-performing variant, Ad-LLM (GIR-R), on the LM-Market dataset for $k \in {1,2,3,4,5}$. **Overall, we observe a clear decrease in response quality as $k$ increases.** In particular, all user-satisfaction metrics (Accuracy, Naturalness, Personality, Trust) decline monotonically under denser ad placement. The Notice metric also decreases, as it captures not only whether users see the injected ads but also their attitude toward noticing them, which becomes more negative when more ads are inserted.
>
> In contrast, the Click metric exhibits a non-monotonic pattern: it initially increases because a higher number of injected ads raises the likelihood that users notice and click at least one link, but it slightly decreases when $k$ becomes larger, as the overall response becomes less relevant and appealing, reducing users’ willingness to click. **This divergence aligns with real-world advertising behavior, where higher ad density increases exposure but diminishes user experience.**
>
> | k | Accuracy  | Naturalness  | Personality  | Trust  | Notice  | Click  | Overall |
> | --- | --- | --- | --- | --- | --- | --- | --- |
> | 1 | 80.32 | 62.49 | 71.88 | 70.47 | 80.14 | 77.10 | 73.73 |
> | 2 | 75.59 | 56.13 | 67.26 | 65.38 | 79.46 | 82.10 | 70.99 |
> | 3 | 72.74 | 48.76 | 65.32 | 60.43 | 77.15 | 80.43 | 67.47 |
> | 4 | 71.72 | 39.19 | 62.31 | 59.41 | 76.29 | 80.86 | 64.96 |
> | 5 | 67.80 | 34.25 | 60.43 | 54.25 | 74.46 | 79.62 | 61.80 |
>
> ### **Response to Q2**
>
> Please see **Common Response #1**.
>
> ### **Response to Q3**
>
> In this work, we assume that the adDB contains factually verified and policy-compliant ads that follow standard platform-level moderation practices (see **Common Response #3**). Therefore, verifying the factual correctness of the ads themselves is orthogonal to our benchmark. We acknowledge that independently evaluating and auditing the factual validity of ad creatives is a critical and promising direction, especially for real-world deployment, and we consider it an important area for future exploration.
>
> ### **Response to Q4**
>
> Please refer to **Common Response #3**.

---

> ### Author Response · Authors · 2025-11-26
>
> Hi, thank you again for taking the time to review our submission. We were wondering if you might have any further comments or feedback. We truly appreciate your efforts and are happy to provide any additional information if needed. Thank you!

---

### Official Review · Reviewer_eZeW · 2025-11-03

**Soundness:** 3
**Presentation:** 3
**Contribution:** 2
**Rating:** 4
**Confidence:** 3

**Summary:**

This paper proposes a benchmark for a newly evolving field of generative engine marketing, whereby ads are generated within an LLM chat history that are supposed to be relevant to user and query. This field is fairly new, and while some models exist for generating such ads, an evaluation is missing.

**Strengths:**

* well written and easy to understand
* GEM-Bench includes two Ad datasets and metrics
* Several LLM judges are tried and yield consistent results

**Weaknesses:**

* the fit to a conference on Learning Representations seems fairly low to me. How best to serve ads - while of course a difficult challenge that industry does encounter - seems to me likely to not be of tremendous interest to the core and broader ICLR community.
* The paper therefore also contains limited novelty as there's no new methodology being proposed, apart from using LLMs as a judge for a new benchmark with existing data
* There are potential problems with ethics: steering LLM behavior and thus users in a "subtle [sic]" way might be easily misused. A clear and thorough discussion on ethics **in the main paper** should be included. I can see how such a paper might also be useful to start the academic study of such ad-systems, so the positioning of this seems critical.
* the core metric of click-through-rate is only simulated based on the LLM judge. Without actual cofirmation of this metrics reliability, the results cannot stand on their own.
* results are only presented with niche LLM of doubao-1.5

* small:
    * Citations should be in \citep and not \cite
    * typos like "EVALUTAION", "Rewritter"

**Questions:**

* How is this paper relevant to a conference on learning representations?
* Method:
** what if you use varying base LLMs to produce ads, e.g. Qwen, llama3, etc.?
** do your simulated human ratings actually correlate with those of real humans? Across what culture and socio-economic groups? It is unlikely that a single LLM judge will be able to simulate many potential different user groups correctly without further tuning. Analysis here would be direly needed.
*

---

> ### Author Response · Authors · 2025-11-20
> **Thanks for Reviewer eZeW’s comments**
>
> ### **Response to W1&Q1**
>
> We thank the reviewer for raising this concern about the fit to ICLR. We understand that the application domain of ads may appear applied at first glance.
> We would like to clarify that we formulate a new problem of ad-injected response generation and set up the experimental environments, including benchmark datasets, initial baselines, and evaluation metrics. This platform serves as **the backbone for developing learning representation related solutions**, for example studying how to perform response generation that balances user satisfaction and engagement as multiple objectives through fine-tuning and alignment, which is central to the study of learning representations at ICLR.
>
> ### **Response to W2**
>
> We thank the reviewer for the comment. We would like to summarize that the main novelty comes from:
>
> 1. The **formulation of Ad-Injected Response (AIR) generation** as a new problem setting;
> 2. The **proposal of a measurement ontology**, which includes the *LLM-as-a-judge* dimensions mentioned by the reviewer;
> 3. The **design of solution Ad-LLM**. Feizi et al. (2023) introduce the first conceptual framework for LLM-based advertising, focusing on outlining the stages of the emerging advertising ecosystem. While they mention the idea of first generating an ad-free response and then inserting suitable ads, their discussion centers on the broader mechanism-design questions, such as whether the platform or the advertiser performs the modification and the associated challenges, without providing a concrete strategy for how the modification should be carried out. In contrast, **Ad-LLM implements a modularized and multi-agent pipeline for AIR generation**, comprising: **(i) raw response generation, (ii) RAG-based ad retrieval, (iii) identifying proper ad-injection positions based on flow-disturbance minimization, and (iv) final response refinement.** In the revised paper, we have moved key steps of Ad-LLM from the appendix to Section 3.3 to better highlight the Ad-LLM design and clarify in the introduction.
>
> ### **Response to W3**
>
> Please refer to **Common Response #3**.
>
> ### **Response to W4**
>
> Please refer to **Common Response #2**.
>
> ### **Response to W5**
>
> We thank the reviewer for this constructive comment. As suggested, in addition to *doubao-1-5-lite*, we further evaluate both the baseline **Ad-Chat** and the best-performing solution **Ad-LLM (GIR-R)** on the **LM-Market** and **CA-Product** datasets across **four additional base LLMs**: **qwen3-30b-a3b, llama3.3-70B, gemini-2.0-flash-lite, gemini-2.5-flash-lite.**
>
> Following the format of Table 5 in the submitted paper, we report the **overall qualitative scores** of different base LLMs in the table below. We observe that **Ad-LLM (GIR-R) consistently outperforms Ad-Chat across all models and both datasets**. In particular, Ad-LLM (GIR-R) achieves an average relative improvement of **~10% on LM-Market** and **~16% on CA-Prod** over Ad-Chat. Furthermore, comparing gemini-2.0-flash-lite and gemini-2.5-flash-lite, we observe that although the superiority of Ad-LLM remains consistent, **stronger base LLMs naturally yield higher absolute generation quality for both solutions**, indicating that AIR generation method benefits from and scales with stronger underlying models.
>
> | **Dataset** | **Solution** | **doubao-1.5-lite** | **llama3.3-70B** | **qwen3-30b-a3b** | **gemini-2.0-flash-lite** | **gemini-2.5-flash-lite** |
> | --- | --- | --- | --- | --- | --- | --- |
> | **LM-Market** | Ad-Chat | 63.81 | 64.95 | 70.91 | 71.32 | 73.17 |
> |  | **Ad-LLM (GIR-R)** | **73.73** | **75.27** | **77.88** | **73.25** | **77.47** |
> | **CA-Prod** | Ad-Chat | 50.51 | 56.42 | 60.08 | 59.38 | 63.42 |
> |  | **Ad-LLM (GIR-R)** | **58.34** | **65.71** | **77.92** | **65.17** | **66.17** |
>
> ### **Response to W6**
>
> We have thoroughly proofread the paper, correct all typos, and use `\citep` as required.
>
> ### **Response to Q1**
>
> Please see **response to W5** and **Common Response #1.**

---

> ### Author Response · Authors · 2025-11-26
>
> Hi, thank you again for taking the time to review our submission. We were wondering if you might have any further comments or feedback. We truly appreciate your efforts and are happy to provide any additional information if needed. Thank you!

---

### Author Response · Authors · 2025-11-20
**Cover Letter**

We thank all reviewers for their insightful and constructive comments, which helped us improve this paper.

**Our response will include addressing common issues raised by the reviewers and providing a point-by-point response to each reviewer's specific concerns. We have uploaded the revised manuscript and highlighted the revisions in blue.**

To summarize, we made the following main updates as advised.

- **GEM-Bench (Section 3):**
    - Section 3.1: We clarified the query counts of the LM-Market datasets in response to Reviewer **tDUn**.
    - Section 3.2: As advised by Reviewer **tDUn**, we included a more detailed explanation of the design and scoring rubric of the LLM judges.
    - Section 3.3: We reorganized the presentation of the proposed framework Ad-LLM to ensure the technical contributions are introduced precisely and concisely, and we adjusted the corresponding paragraph in Section 1 in response to Reviewer **eZeW** and **hUJt**.
- **Experiments (Section 4):** We have included all experiments mentioned by the reviewers.
    - Section 4.1: As suggested by Reviewer **eZeW**, we included the performance under 4 additional base LLMs. We also added a set of experiments to evaluate the quality of ad-injected responses with different numbers of inserted ads, as advised by Reviewer **tDUn**.
    - Section 4.2: As suggested by all reviewers (**eZeW**, **tDUn**, and **hUJt-W1**), we added an additional user study section to demonstrate human–LLM alignment for all qualitative metrics.
    - Configuration and appendices: For the experiments on query size (Reviewer **tDUn**) and different system prompts (Reviewer **hUJt**), due to space limitations, we included the main conclusions in the experiment setup paragraph and placed the detailed results in the appendices.
- **Ethics Statement (Section 6):** As pointed out by Reviewer **eZeW** and **tDUn**, we updated the ethics statement to further discuss potential ethical risks and the responsible deployment of ad-injected response generation.

We look forward to hearing your thoughts on our responses. If there are any misunderstandings or areas that require further clarification, we would appreciate any further feedback.

---

> ### Author Response · Authors · 2025-11-20
> **Common Response #1: Human Evaluation**
>
> We thank all reviewers (**eZeW-Q2**, **tDUn-W2/Q2**, **hUJt-W1**) for the helpful suggestions regarding human evaluation. Following their advice, we conducted a comprehensive user study to manually evaluate ad-injected responses across six qualitative dimensions.
>
> ### **User Study Configuration**
>
> We recruited 118 participants with diverse demographic backgrounds (details in the revised paper). They evaluated responses from **Ad-Chat and Ad-LLM (GIR-R)** along one of six qualitative dimensions **(accuracy, naturalness, personality, trust, notice, and click)**. Ratings followed a four-point Likert scale using the same dimension-specific descriptions as in our LLM-as-a-judge setup to ensure comparability. We selected **20 queries via stratified sampling across three datasets.** To avoid fatigue, each participant rated only 10 randomly sampled queries under a single qualitative dimension, with two model outputs shown side by side in randomized order. Each judgment yields a tuple of (query, method, dimension, score).
> Since some participants contributed multiple submissions evaluating different qualitative dimensions, we obtained **157 valid questionnaires and 3,140 ratings** at last. Each of the 40 (query, method) items received on average 65 independent human judgments, providing sufficient data for reliable split-half consistency estimation and stable correlation analysis with LLM-as-a-judge scores.
>
> ### **Human Reliability**
>
> We first evaluate the internal consistency of human annotators. We compute the split-half Spearman correlation $\rho_{hh}$ based on the overall qualitative score. Participants are randomly divided into two groups, and for each group, we calculate the mean overall score for every (question, method) item by averaging all corresponding ratings. This produces two item-level score vectors covering all 40 items. We then compute the Spearman correlation between the two vectors. The random split is repeated 100 times, and $\rho_{hh}$ is reported as the average correlation across runs. **Overall, human reliability is $\rho_{hh} = 0.84$, indicating strong agreement between independent groups of annotators.**
>
> ### **Human–LLM Alignment**
>
> Following the same procedure, we quantify the agreement between the LLM-as-a-judge and human judgments using the Spearman correlation $\rho_{hl}$. We use the default judge LLM, **\texttt{gpt-4.1-mini}**, to compute the item-level score vector for the 40 items. The human-side vector is obtained by averaging all ratings across valid questionnaires. **We find that the corresponding Spearman correlation is $\rho_{hl} = 0.73$, indicating strong alignment between human and LLM judgments.**
>
> ### **Human-LLM Preference**
>
> To further examine the alignment between human and LLM-based evaluations, we analyze whether both sources favor the same method across the six qualitative metrics. **As shown in the table below, both human raters and the LLM judge consistently assign higher scores to GIR-R than to Ad-Chat on nearly all dimensions.** GIR-R outperforms Ad-Chat in Accuracy, Naturalness, Personality, Trust, and Notice, with an even larger margin under LLM judgments. This consistent pattern highlights a shared preference for GIR-R and demonstrates strong alignment between human and LLM assessments.
>
> | Source | Method | Accuracy | Naturalness | Personality | Trust | Notice | Click |
> | --- | --- | --- | --- | --- | --- | --- | --- |
> | Human | Ad-Chat | 51.41 | 52.50 | 44.90 | 44.56 | 42.00 | 69.36 |
> | Human | GIR-R | 69.31 | 66.14 | 60.10 | 57.22 | 59.52 | 69.00 |
> | LLM | Ad-Chat | 43.50 | 45.00 | 39.00 | 30.00 | 67.50 | 79.50 |
> | LLM | GIR-R | 84.00 | 51.00 | 69.00 | 60.00 | 81.00 | 79.50 |

---

> ### Author Response · Authors · 2025-11-20
> **Other Common Responses**
>
> ### **Common Response #2: Click Measurement**
>
>
> We thank the reviewers **eZeW** and **hUJt** for pointing this out and apologize for the confusion.
>
> Our work involves **two distinct click-related measures**:
>
> 1. **CTR**, which is derived from **ground-truth user click data** in the Microsoft AdsCVLR dataset. This dataset contains real user interaction logs that link queries and click behaviors to relevant ads. Each query is associated with ads displayed on the results page, along with binary click labels (clicked vs. unclicked).
> 2. **Click**, which is a **simulated measure generated by the LLM judge**. To validate the reliability of this simulated metric, we conducted an additional user study, providing a **human-grounded reference for click behavior** and demonstrating **strong alignment between user judgments and LLM-based evaluations** (see **Common Response #1**).
>
>
> ### **Common Response #3: Ethics Statements**
>
>
> We fully agree with the reviewers’ concerns regarding ethical risks and the responsible deployment of ad-injected response generation.
>
> In response to **Reviewer eZeW**, our work is positioned from the **platform’s perspective**, aiming to support responsible research on ad injection rather than promote manipulation. Preventing misuse should rely on **transparent ad-disclosure practices** implemented at the platform level, analogous to the *“Sponsored”* tag used in traditional search-engine advertising. Furthermore, the **specific forms of disclosure and user acceptance** represent an important direction for future research, requiring close collaboration between academia and industry, as well as joint efforts across **AI and business disciplines**. We also believe this area calls for **updated advertising regulations and policy frameworks** to ensure fairness and accountability in generative systems.
>
> In response to **Reviewer tDUn**, from the same platform-centered perspective, **ad safety control** is a fundamental responsibility. Our framework assumes that ads originate from a **verified and policy-compliant ad database** curated by the platform. To prevent deceptive or misleading ads, **content moderation and authenticity screening** should be applied before both the bidding and retrieval stages. Additionally, during the **bidding and auction** process, mechanisms such as **quality-based bidding scores,** which combine relevance, trust, and historical reliability (see Refs. [1–3]), can be used to **down-rank or exclude low-trust advertisers**, ensuring that only credible ads are eligible for injection.
>
> We will integrate these discussions and the corresponding details into the **Ethics Statement (Section 6)** of the revised manuscript to make our ethical stance and platform-level safeguards more explicit.
>
>
> [1] Zhang, H., Chau, D. H., Faloutsos, C., Wang, H., & Faloutsos, M. (2011). Detecting adversarial advertisements in the wild. In Proceedings of the 17th ACM SIGKDD International Conference on Knowledge Discovery and Data Mining (KDD ’11).
>
> [2] Yuan, Y., Wang, F.-Y., & Wang, J. (2014). Reputation-aware continuous double auction. In Proceedings of the 28th AAAI Conference on Artificial Intelligence (AAAI 2014).
>
> [3] Rahme, J., Zheng, L., & Zhang, S. (2021). Auction learning as a two-player game. In Proceedings of the 9th International Conference on Learning Representations (ICLR 2021).

---

### Meta-Review · Area_Chair_ViSZ · 2026-01-11

**Summary:**

The paper introduces a benchmark for evaluating ad-injected response (AIR) generation in both chatbot and search settings. Two of the reviewers are mildly negative and one of the reviewers is mildly positive, and they highlight a number of concerns. Most prominently, the significance of the contribution to ICLR as well as the absence of human input for the creation of the benchmark, relying entire on LLM-as-a-judge. As a result, I am unable to recommend acceptance.

**Reviewer Concerns:**

I do not think the main concerns regarding ICLR fit and use of LLM as a judge have been addressed by the rebuttal.

**Reviewer Scores:**

I do not think the reviewers would have changed their score.

---

### Decision · Program_Chairs · 2026-01-26

Reject